# A nigro–subthalamo–parabrachial pathway modulates pain-like behaviors

Tao Jia[1], Ying-Di Wang[1], Jing Chen[1], Xue Zhang[1], Jun-Li Cao [1,2,3] ✉, Cheng Xiao [1,2,3] ✉ & Chunyi Zhou [1,2,3] ✉

The basal ganglia including the subthalamic nucleus (STN) and substantia nigra pars reticulata (SNr) are involved in pain-related responses, but how they regulate pain processing remains unknown. Here, we identify a pathway, consisting of GABAergic neurons in the SNr (SNr$^{GABA}$) and glutamatergic neurons in the STN (STN$^{Glu}$) and the lateral parabrachial nucleus (LPB$^{Glu}$), that modulates acute and persistent pain states in both male and female mice. The activity of STN neurons was enhanced in acute and persistent pain states. This enhancement was accompanied by hypoactivity in SNr$^{GABA}$ neurons and strengthening of the STN–LPB glutamatergic projection. Reversing the dysfunction in the SNr$^{GABA}$-STN$^{Glu}$-LPB$^{Glu}$ pathway attenuated activity of LPB$^{Glu}$ neurons and mitigated pain-like behaviors. Therefore, the SNr$^{GABA}$-STN$^{Glu}$-LPB$^{Glu}$ pathway regulates pathological pain and is a potential target for pain management.

Pain is a common source of distress for both outpatients and inpatients[1–3]. In spite of its high prevalence, clinical management of pain remains difficult, especially for chronic pain[4–7]. One reason for this is that chronic pain has a multifaceted pathophysiology including central sensitization, which involves plasticity in multiple segments of ascending and descending pain pathways[5,8–14]. Drug development has focused on disrupting such pain-related plasticity. However, recent studies have shown that many nuclei outside the classical pain pathways dramatically modulate pain processing[15–20]. Dissection of these circuits may provide potential therapeutic targets for effective treatment of chronic pain.

The subthalamic nucleus (STN) is the mere glutamatergic nucleus in the basal ganglia and modulates motor, limbic, and cognitive functions[21–24]. Studies in humans and rodents have revealed that the STN may play a significant role in pain perception and modulation. For instance, STN neurons have fast spontaneous firing and readily respond to painful stimuli[25,26]. Optogenetic activation of STN neurons decreases pain thresholds in mice[20]. Consistent with previous studies showing that deep brain stimulation (DBS) in the STN relieves pain in patients with Parkinson's disease[27–33], we demonstrated that optogenetic inhibition of hyperactive STN neurons ameliorates hyperalgesia and central sensitization in parkinsonian mice[20]. The STN sends glutamatergic projections to both basal ganglia and non-basal ganglia structures[21,23]. However, it remains an open question: whether there is a connection through which hyperactive STN neurons lead to the sensitization of pain pathways underlying chronic pain. Given that the lateral parabrachial nucleus (LPB) is a major component in the pain signaling pathway[34] and is one of nuclei activated following optogenetic stimulation of the STN[20], we postulated that STN neurons may modulate activity of the LPB and thus pain processing.

The STN receives excitatory and inhibitory inputs from various cortical and subcortical areas[21,23,35,36]. Our recent study found that activation of GABAergic inputs from the substantia nigra pars reticulata (SNr) to the STN reduced pain-like hypersensitivity during inflammatory and neuropathic pain states[37]. In particular, neuronal activity in the SNr was compromised under neuropathic pain condition, indicating a role of GABAergic inputs in the STN in the modulation of pain signals. Besides the SNr, the STN receives dense GABAergic inputs from the globus pallidus externa (GPe)[21]. It remains unknown whether these inhibitory inputs to STN neurons are modified under

[1]Jiangsu Province Key Laboratory in Anesthesiology, School of Anesthesiology, Xuzhou Medical University, 221004 Xuzhou, Jiangsu, China. [2]Jiangsu Province Key Laboratory of Anesthesia and Analgesia Application Technology, Xuzhou Medical University, 221004 Xuzhou, Jiangsu, China. [3]NMPA Key Laboratory for Research and Evaluation of Narcotic and Psychotropic Drugs, School of Anesthesiology, Xuzhou Medical University, 221004 Xuzhou, Jiangsu, China. ✉e-mail: caojl0310@aliyun.com; xchengxj@xzhmu.edu.cn; chunyi.zhou@xzhmu.edu.cn

neuropathic pain condition, affecting STN neuronal activity and the processing of neuropathic pain signals.

In the present study, we combined viral tracing, electrophysiology, optogenetics, chemogenetics, fiber photometry, and pain-like behavior assays to verify the functional organization of the SNr–STN–LPB pathway and explore the role of this pathway in modulating acute and persistent pain states. Furthermore, we interrogated dysregulation of the SNr–STN–LPB pathway in different pain states. We demonstrate that selective manipulation of the tandem pathway (SNr[GABA] to STN[Glu] to LPB[Glu]) ameliorates pain-like behaviors in several pain states.

## Results

### STN neurons are activated in acute and persistent pain states

To reveal the role of STN neurons in acute and persistent pain states, we examined the expression of c-Fos (a marker of enhanced neural activity) in the STN after mice were subjected to unilateral injection of capsaicin in the lower hind leg or complete Freund's adjuvant (CFA) in the hind paw or spared nerve injury (SNI) (Fig. 1a, b, f, j). Successful establishment of these pain models was confirmed by reduced mechanical paw withdrawal threshold (PWT) and shortened thermal paw withdrawal latency (PWL) (Fig. 1c, g, k). We observed a dramatic increase in the number of c-Fos-positive neurons in the bilateral STN in the unilateral capsaicin (1 h, Fig. 1d, e), CFA (1 day, Fig. 1h, i), and SNI (week 2 after SNI; Fig. 1l, m) pain models. We also observed an increase of c-Fos expression in the STN 3 weeks and 5 weeks after SNI surgery (Fig. 1n–q), suggesting that pain-like behaviors are concomitant with alteration in the STN. Consistent with these findings, STN neurons exhibited a higher evoked firing rate in brain slices from SNI mice than in slices from sham mice (1–2 weeks after SNI surgery; Fig. 1r, s). However, no differences were observed in the resting membrane potential or the input resistance of STN neurons between sham and SNI mice (Fig. 1t, u, Supplementary Table 1). These data support the hypothesis that STN neurons are hyperactive in acute and persistent pain states, in addition to parkinsonian pain[20].

To investigate changes in the activity of STN neurons associated with pain-like hypersensitivity, we intracranially injected AAV-CaMKII-GCaMP6f into the STN and recorded the GCaMP6f signal from STN neurons in mice anesthetized with 1.0% isoflurane to avoid movement artifacts from withdrawal behavior (Fig. 2a, b; Supplementary Fig. 1a, b). We detected a transient increase in GCaMP6f signal following a strong mechanical stimulus (4 g von Frey filament) or a heat (48 °C) stimulus of either hind paw (Fig. 2c–r). To ascertain how STN neurons respond to external stimulus in inflammatory and neuropathic pain states, we recorded GCaMP6f signal in STN neurons and stimulated hind paws with a von Frey filament corresponding to PWT in mice receiving capsaicin or saline injection in lower hind legs or SNI (0.16 g, 0.6 g, and 0.16 g, respectively). We found that the threshold stimulation elicited similar increase of GCaMP6f signal in mice 30–60 min after capsaicin injection in lower hind legs or 1–2 weeks after SNI surgery (Fig. 2s–x), whereas a smaller increase of GCaMP6f signal in mice that saline was injected in hind legs (Fig. 2u, x). Given that the CaMKII promoter drives gene expression in glutamatergic neurons in the STN (STN[Glu])[38], these results suggest that activation of STN[Glu] neurons are associated with pain-like hypersensitivity in pain states.

### STN[Glu] neurons directly innervate LPB[Glu] neurons

Our previous study demonstrated that stimulation of STN neurons reduces both the mechanical PWT and the thermal PWL by promoting central sensitization[20]. However, it is unknown whether these neurons regulate ascending or descending pain pathways. To dissect the circuit underlying the involvement of the STN in pain-like responses, we mapped the downstream targets of the STN. We injected Alexa 488-conjugated WGA, a trans-synaptic tracer[39], into the STN and observed

fluorescently labeled neurons in a number of brain regions, including the LPB (Fig. 3a). Because the principal glutamatergic neurons in the LPB (LPB[Glu]) play important roles in pain processing[14,40,41], we postulated that the LPB neuronal types innervated by the STN may be glutamatergic. To prove that, we injected AAV-EF1α-DIO-eYFP into the LPB in Vglut2-Cre mice to label LPB[Glu] neurons and WGA-Alexa 555 into the STN to label LPB neurons receiving the STN projection (Fig. 3b, c). We observed nearly 80% of WGA-Alexa 555(+) neurons in the LPB were co-labeled with eYFP (Fig. 3c). To further confirm STN - LPB[Glu] projection, we injected a trans-synaptic anterograde viral vector, scAAV1-hSyn-Cre, into the STN of Ai9 mice (a tdTomato reporter line) (Fig. 3d, e). After 3 weeks' recovery, numerous tdTomato(+) cells were observed in the LPB (Fig. 3f) and about 75% of these cells co-expressed CaMKII, a marker for glutamatergic neurons (Fig. 3g). To characterize the functionality of the STN–LPB projection, we injected AAV-CaMKII-ChR2-eYFP into the STN and observed that blue light stimulation (20 Hz, 5 ms) elicited CNQX-sensitive excitatory postsynaptic currents (photo-EPSCs) in 71% (17 of 25) of LPB neurons surrounded by abundant ChR2-labeled nerve terminals (Fig. 3h, i, j, k). In these responsive neurons, the firing rate was significantly increased by photo-stimulation (10 Hz and 20 Hz) of the STN terminals in the LPB, and the effect of photo-stimulation was more robust at 20 Hz than that at 10 Hz (Fig. 3l, m).

Retrograde tracing[42] from the LPB also indicated the existence of monosynaptic projections from the STN (Supplementary Fig. 2a–c). Quantification of retrograde labeling from the LPB showed that about 32.62% of neurons in the STN projected to the LPB (Fig. 3n–p), suggesting that this pathway may play an important contribution to LPB modulation of the pain signals. These results indicate that the majority of LPB neurons innervated by the STN are glutamatergic.

Our data indicate that the STN–LPB projection is glutamatergic, consistent with the fact that a great majority of STN neurons are glutamatergic neurons[23]. In line with the slice recording data, ChR2-mediated activation of STN terminals increased the number of c-Fos-positive neurons in the LPB in freely moving mice (Supplementary Fig. 2h, i). These experiments demonstrate that optogenetic stimulation of STN glutamatergic terminals is sufficient to increase the activity of LPB neurons.

### The role of STN–LPB neurons in modulation of pain threshold in different pain states

To explore the contribution of LPB-projecting STN neurons (STN–LPB neurons) to pain processing, we expressed ChR2 or NpHR in STN–LPB neurons, by injecting AAV-CaMKII-Cre-eGFP in the STN in combination with the injection of AAV-retro-DIO-ChR2-mCherry or AAV-retro-DIO-NpHR-mCherry into the ipsilateral LPB (Fig. 4a; Supplementary Fig. 3a, b, e, f). This strategy allows for efficient manipulation of STN–LPB neurons, as ascertained by our brain slice patch-clamp recording data (Supplementary Fig. 3c, d, g, h). Optogenetic activation (20 Hz, 5 ms pulse, 473 nm laser) of the STN–LPB neurons induced bilateral mechanical hypersensitivity (Fig. 4b), but did not change the thermal PWL (Fig. 4c), whereas optogenetic inhibition (continuous pulse, 589 nm laser) of STN–LPB neurons had no effect on mechanical or thermal thresholds in either hind paw (Fig. 4d, e). Neither blue nor yellow-light illumination of the STN regulated pain thresholds in control mCherry mice (Fig. 4b–e). Additionally, motor performance in the open field test was not affected by blue or yellow-light illumination of the STN in ChR2, NpHR, or mCherry mice (Supplementary Fig. 3l–q). These data support the notion that stimulation of STN–LPB neurons induces mechanical hypersensitivity in naïve mice.

Our finding that excitation of STN–LPB neurons is sufficient to induce mechanical hypersensitivity, mimicking pain states, prompted us to investigate whether inactivation of these neurons ameliorates pain-like hypersensitivity in inflammatory and neuropathic pain states. We used the same combinatorial optogenetic strategy shown in Fig. 4a to selectively transfect NpHR into STN–LPB neurons. Unilateral

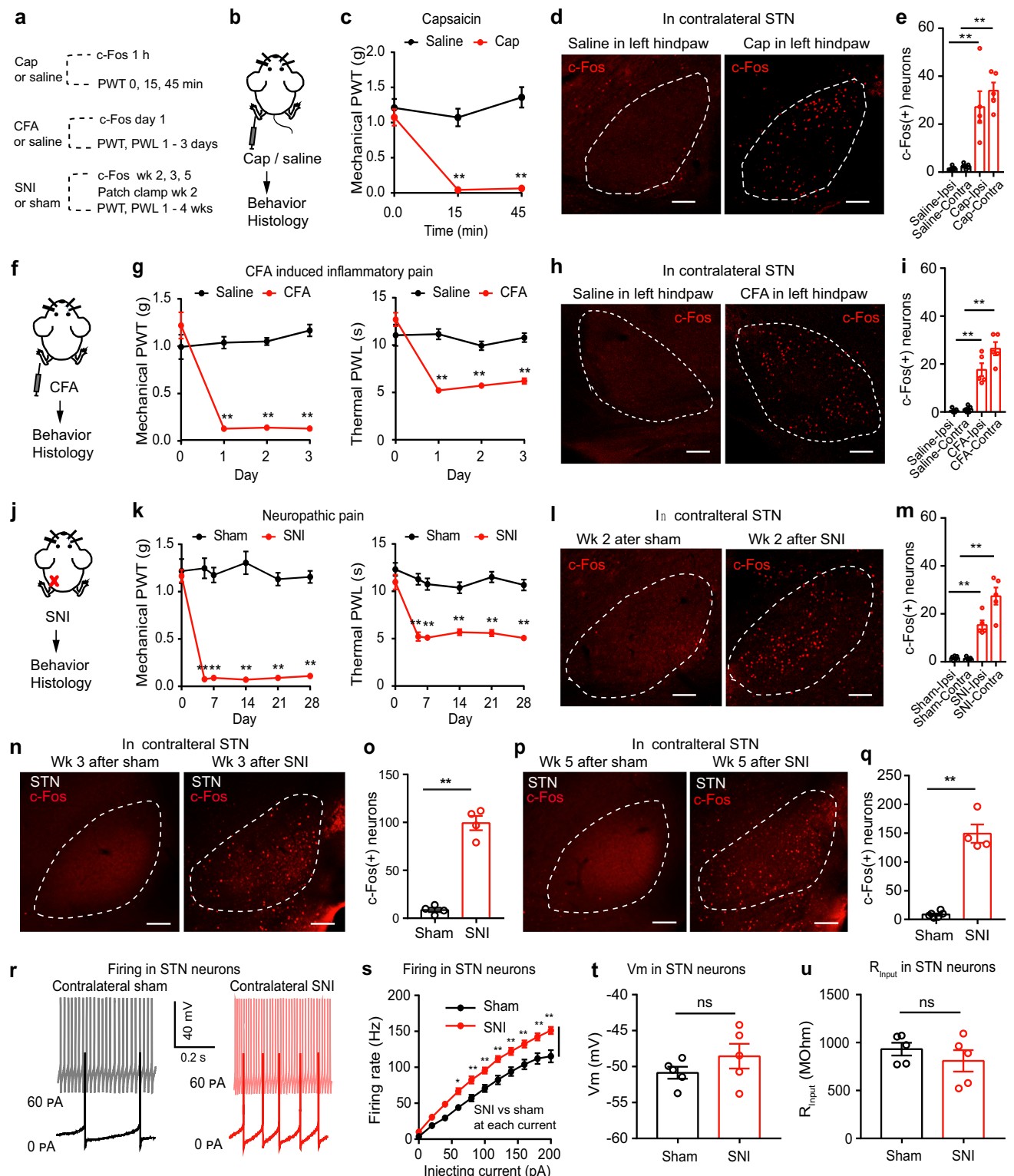

**Fig. 1 | STN neurons are activated in pain-like behaviors. a** Experimental diagram. **b, c** Pain development evoked by capsaicin. **c** PWT: $F_{(2, 36)} = 16.62$, $P < 0.001$; $n = 7$ per group. **d, e** Example images (**d**) and quantification (**e**) of c-Fos-positive neuron after saline or capsaicin (Cap). $F_{(3, 16)} = 20.09$, $P < 0.0001$; $n = 5$ per group. **f, g** Pain development after CFA injection. PWT: $F_{(3, 44)} = 29.31$, $P < 0.001$; PWL: $F_{(3, 44)} = 17.67$, $P < 0.0001$; $n \geq 6$ per group. **h, i** Example images (**h**) and quantification (**i**) of c-Fos-positive neurons after saline or CFA. $F_{(3, 16)} = 42.17$, $P < 0.0001$; $n = 5$ per group. **j, k** Pain development after SNI. **k** PWT: $F_{(5, 78)} = 21.23$, $P < 0.0001$; PWL: $F_{(5, 78)} = 6.12$, $P < 0.0001$; $n \geq 7$ per group. **l, m** Example images (**l**) and quantification (**m**) of c-Fos-positive neurons in week 2 after sham or SNI ($F_{(3, 16)} = 37.97$, $P < 0.0001$; $n = 5$ per

group). **n–q** Example images (**n, p**) and quantification (**o, q**) of c-Fos-positive neurons after sham or SNI on the contralateral side ($n \geq 4$ per group). **o** $t = 11.56$, $P < 0.0001$; **q** $t = 10.94$, $P < 0.0001$. **r–u** Example traces (**r**), evoked firing (**s**), resting membrane potential ($V_m$) (**t**) and input resistance ($R_{input}$) (**u**) in contralateral STN neurons in week 2 after sham or SNI. **s** $F_{(1, 214)} = 128.9$, $P < 0.0001$, $n = 4$ per group. **t** $t = 1.21$, $P = 0.26$, $n = 5$ per group. **u** $t = 0.96$, $P = 0.37$. $n = 5$ per group. **\*\*** $P < 0.01$; Two-way ANOVA with Tukey's post-hoc analysis for (**c, g, k** and **s**); One-way ANOVA with Tukey's post-hoc analysis for (**e, i** and **m**); Two-tailed unpaired $t$ test for (**o, q, t,** and **u**). Data are presented as mean ± SEM. Scale bars: 100 μm.

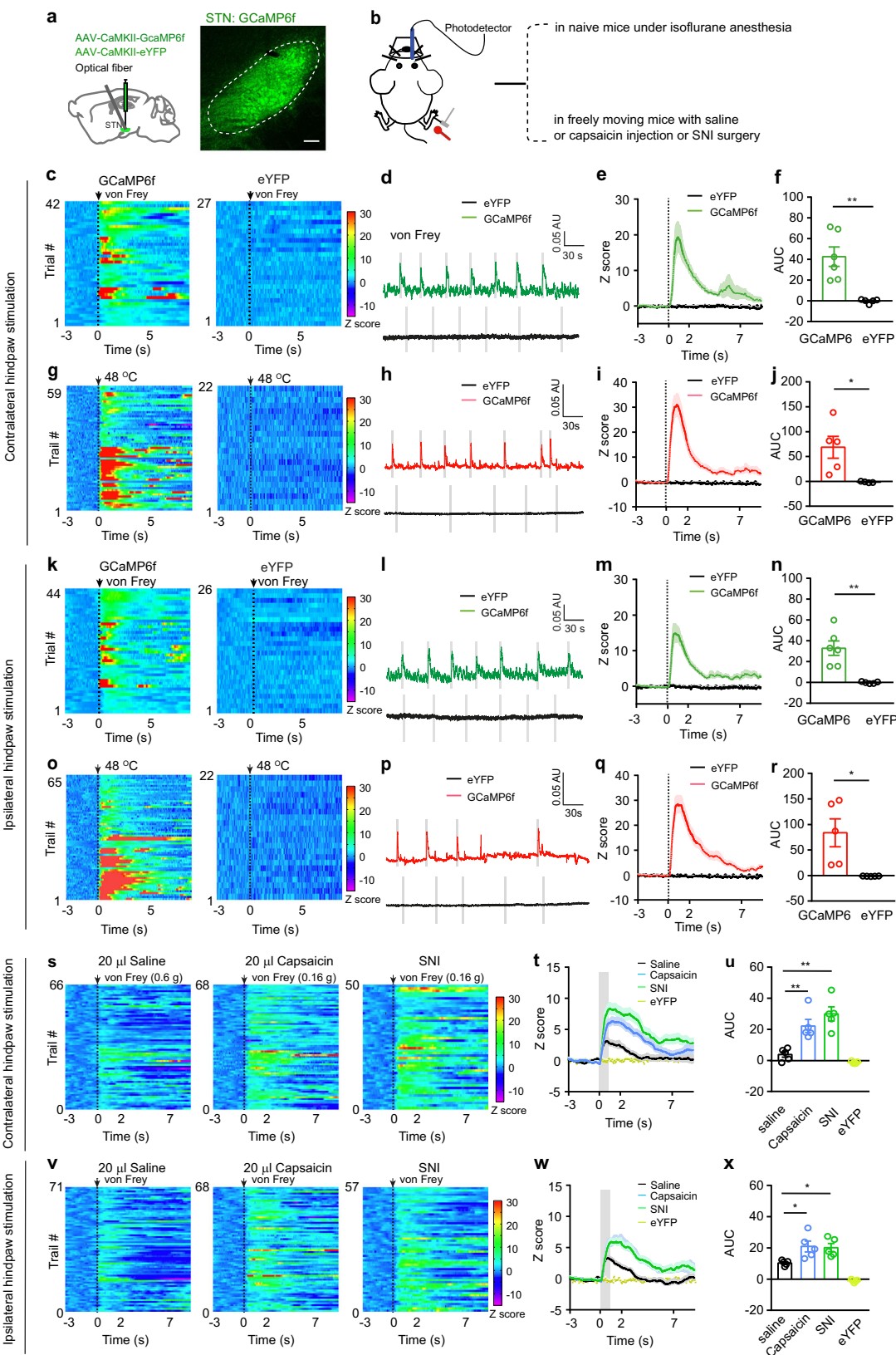

optogenetic silencing of these STN neurons with yellow-light preferentially suppressed capsaicin-induced acute nocifensive behaviors (such as hind-paw licking and flinching) for the contralateral but not the ipsilateral hind paw (Fig. 4f, g; Supplementary Fig. 4a). As sensory information is sent from the dorsal horn of the spinal cord to the contralateral hemisphere through an ascending pathway[43–45], these

results indicate that STN–LPB neurons may regulate the ascending pain pathway. Since subcutaneous injection of capsaicin leads to delayed but longer-lasting secondary mechanical allodynia in adjacent tissue due to central sensitization[46], we measured hind paw mechanical threshold during the early (15–30 min) and late (45–60 min) phases following capsaicin injection in the lower hind leg. We found that

**Fig. 2 | Dynamics of STN GCaMP6 signal in pain processing. a** Experimental diagram and example image of GCaMP6f in the STN. **b** Experimental diagram for GCaMP6 recordings. **c–j** Heat maps (**c**, **g**), example traces (**d**, **h**), averaged normalized traces (**e**, **i**), and quantification (**f**, **j**, area under the curves (AUC) in panels **e** and **i**) of changes in GCaMP6f and eYFP signals in mice receiving von Frey or heat stimulation on the contralateral hind paw. **f** $t = 4.12$, $p = 0.0026$, $n \geq 5$ per group; **j** $t = 3.16$, $p = 0.014$, $n = 5$ per group. **k–r** Heat maps (**k**, **o**), example traces (**l**, **p**), averaged normalized traces (**m**, **q**), and quantification (**n** and **r**, AUC in panels **m** and **q**) of changes in GCaMP6f and eYFP signals in mice receiving von Frey or heat stimulation on the ipsilateral hind paw. **n** $t = 4.33$, $p = 0.0019$, $n \geq 5$ per group. **r** $t = 3.12$, $p = 0.014$, $n = 5$ per group. **s–x** Heat maps (**s**, **v**), averaged normalized traces (**t**, **w**), and quantification (**u** and **x**, AUC in panels **t** and **w**) of changes in GCaMP6f and eYFP signals following von Frey stimulation of the contralateral and ipsilateral hind paw in mice receiving saline or capsaicin injection in lower hind leg (30 min) or SNI surgery (2 weeks). **u** $F_{(3, 16)} = 21.13$, $p < 0.0001$. **x** $F_{(3, 15)} = 21.6$, $p < 0.0001$. $n = 5$ per group. **\*\*$P < 0.01$**; Two-tailed unpaired $t$ test for (**f**, **j**, **n**, and **r**); One-way ANOVA with Tukey's post-hoc analysis for (**u**) and (**x**). Data are presented as mean ± SEM. AU in panels (**d**, **h**, **l** and **p**) stands for arbitrary unit of fluorescence intensity. Scale bar: 100 μm.

unilateral optogenetic silencing of STN–LPB neurons mitigated the capsaicin-induced reduction of the mechanical pain threshold for the contralateral but not the ipsilateral hind paw (Fig. 4h, i; Supplementary Fig. 4b). In persistent pain states, such as CFA-induced inflammatory pain (Fig. 4j, k; Supplementary Fig. 4c) and SNI-induced neuropathic pain (Fig. 4l–n), we also observed that optogenetic silencing of STN–LPB neurons reduced mechanical and thermal hypersensitivity of the contralateral hind paw. Therefore, the STN–LPB projection regulates not only acute pain but also persistent pain that involves central sensitization.

Pain-relieving effect of optogenetic inhibition of STN–LPB neurons was further examined in the conditioned place preference (CPP) test (Fig. 4l), an operant behavioral test based on the fact that pain-relieving treatments are rewarding and produce a CPP in mice with nerve injury[47–49]. SNI mice with NpHR expression in STN–LPB neurons showed a preference to the yellow-light conditioned chamber (Fig. 4o, p). Such preference was not seen in sham mice or SNI mice with mCherry expression in STN–LPB neurons (Fig. 4o, p, r, s). Our observation that light stimulation did not reduce the traveling velocity in the light stimulation-paired chamber (Fig. 4q, t) excludes the possibility that altered locomotion interfered with the latency to shuttle between chambers. Optogenetic silencing of STN–LPB neurons also attenuated the elevation of c-Fos-positive STN neurons in capsaicin, CFA, and SNI mice (Supplementary Fig. 4d–f).

Taken together, these results indicate that STN–LPB neurons modulate pain-like hypersensitivity.

## The role of the STN^Glu-LPB^Glu projection in pain-like hypersensitivity

STN–LPB neurons also send axonal outputs to other nuclei besides the LPB and any such divergent projections may contaminate the behavioral results following stimulation of STN–LPB neurons. We first mapped the axonal outputs of STN–LPB neurons using the same virus strategy shown in Supplementary Fig. 3a. We observed STN–LPB axonal fibers in the pedunculopontine nucleus (PPN), substantia nigra reticulata (SNr), substantia nigra compacta (SNc), ventral tegmental area (VTA), globus pallidus interna and externa (GPi and GPe) and ventral pallidum (VP) besides the LPB (Supplementary Fig. 3i, j). Among these, the LPB is the densely innervated area of STN–LPB neurons (Supplementary Fig. 3i, j, k). To test whether the STN–LPB glutamatergic projection is involved in pain-related response in LPB neurons, we injected AAV-CaMKII-hM4Di-mCherry into the STN and Cre-dependent GCaMP6f (AAV-DIO-GCaMP6f) into the LPB of Vglut2-Cre mice (Fig. 5a, b; Supplementary Fig. 11a–c). In vivo fiber photometry recordings revealed that the GCaMP6f signal in the LPB was significantly increased following von Frey filament stimulation of the contralateral hind paw, and a mild increase in the GCaMP6f signal was also detected following the same strength of stimulation applied to the ipsilateral hind paw (Fig. 5c–e, g–i). This is in agreement with previous studies showing that the LPB receives a less dense sensory input from ipsilateral than contralateral spinal cord dorsal horn[10,50,51]. Continuous silencing of STN^Glu neurons by the hM4Di agonist, clozapine-N-oxide (CNO; 3 mg/kg, i.p.) significantly attenuated GCaMP6f signal following contralateral but not ipsilateral hind paw stimulation (Fig. 5f, j). These data therefore suggest that disrupting neural activity in the STN

attenuates responses of LPB^Glu neurons to mechanical stimulation on the contralateral hind paw.

Furthermore, we investigated whether STN inputs to the LPB modulate pain thresholds (Fig. 6; Supplementary Fig. 5). WT mice were injected with AAV-CaMKII-ChR2-eYFP, AAV-CaMKII-NpHR-eYFP, or AAV-CaMKII-eYFP in the STN and optical fibers were placed over the LPB (Fig. 6a, b; Supplementary Fig. 5a–c). We observed that optogenetic activation (20 Hz, 5 ms pulse, 473 nm laser) of the STN–LPB projection induced mechanical and thermal hypersensitivity on the contralateral but not the ipsilateral hind paw (Fig. 6c). However, neither the mechanical nor the thermal pain threshold on either hind paw was altered by optogenetic inhibition of the STN–LPB projection (continuous light, 589 nm laser; Fig. 6d). Motor performance in the open field test or adhesive tape removal test was not affected by blue or yellow-light illumination of the LPB in ChR2 or NpHR mice (Supplementary Fig. 5d–g). These results suggest that activation of the STN–LPB projection is sufficient to reduce the pain thresholds, but this projection plays a limited role in maintenance of the normal pain thresholds.

We then interrogated the role of the STN–LPB projection under inflammatory and neuropathic pain conditions. We observed that unilateral optogenetic silencing of the STN–LPB projection significantly mitigated capsaicin-evoked acute nocifensive behaviors in the contralateral but not the ipsilateral hind paw (Fig. 6e, f; Supplementary Fig. 5h). Similarly, contralateral capsaicin-, CFA-, and SNI-induced pain-like hypersensitivity in the hind paw were attenuated by optogenetic silencing of the STN–LPB projection (Fig. 6g–m; Supplementary Fig. 5i). Consistent with this, SNI-induced increase in c-Fos expression in the LPB was markedly reduced by optogenetic silencing of the STN–LPB projection (Supplementary Fig. 5j). These results confirm our observations in the same pain models with optogenetic inhibition of the STN–LPB neurons (Fig. 4).

We also found that the STN–LPB pathway regulates pain-like behaviors in female mice under different pain conditions (Fig. 7) similarly to its effects on pain-like behaviors in male mice (Fig. 6 and Supplementary Fig. 5). We observed that unilateral optogenetic stimulation of the STN–LPB pathway decreased both mechanical and thermal pain thresholds on the contralateral but not the ipsilateral hind paw in naïve female mice (Fig. 7a–e); in contrast, optogenetic inhibition of this pathway had no effect on mechanical and thermal thresholds on both hind paws in naïve female mice (Fig. 7f, g). In capsaicin-, CFA-, and SNI female mice, optogenetic inhibition of the STN–LPB pathway mitigated pain-like behaviors on the contralateral but not the ipsilateral hind paw (Fig. 7h–n). CPP was developed in SNI female mice by optogenetic inhibition of this pathway (Fig. 7o–q). These results indicate that the effects of the STN–LPB pathway on pain-like behaviors may not have sex-differences.

Therefore, our data support that STN–LPB pathway is important for modulating pain-like hypersensitivity. Further studies are needed to explore the roles of the synaptic connections between the STN and other outputs in pain-like hypersensitivity.

## The STN^Glu-LPB^Glu projection is modified in neuropathic pain

Persistent pain has been associated with changes in the plasticity of the pain transmission pathways[9,12]. The ability of STN inhibition to relieve

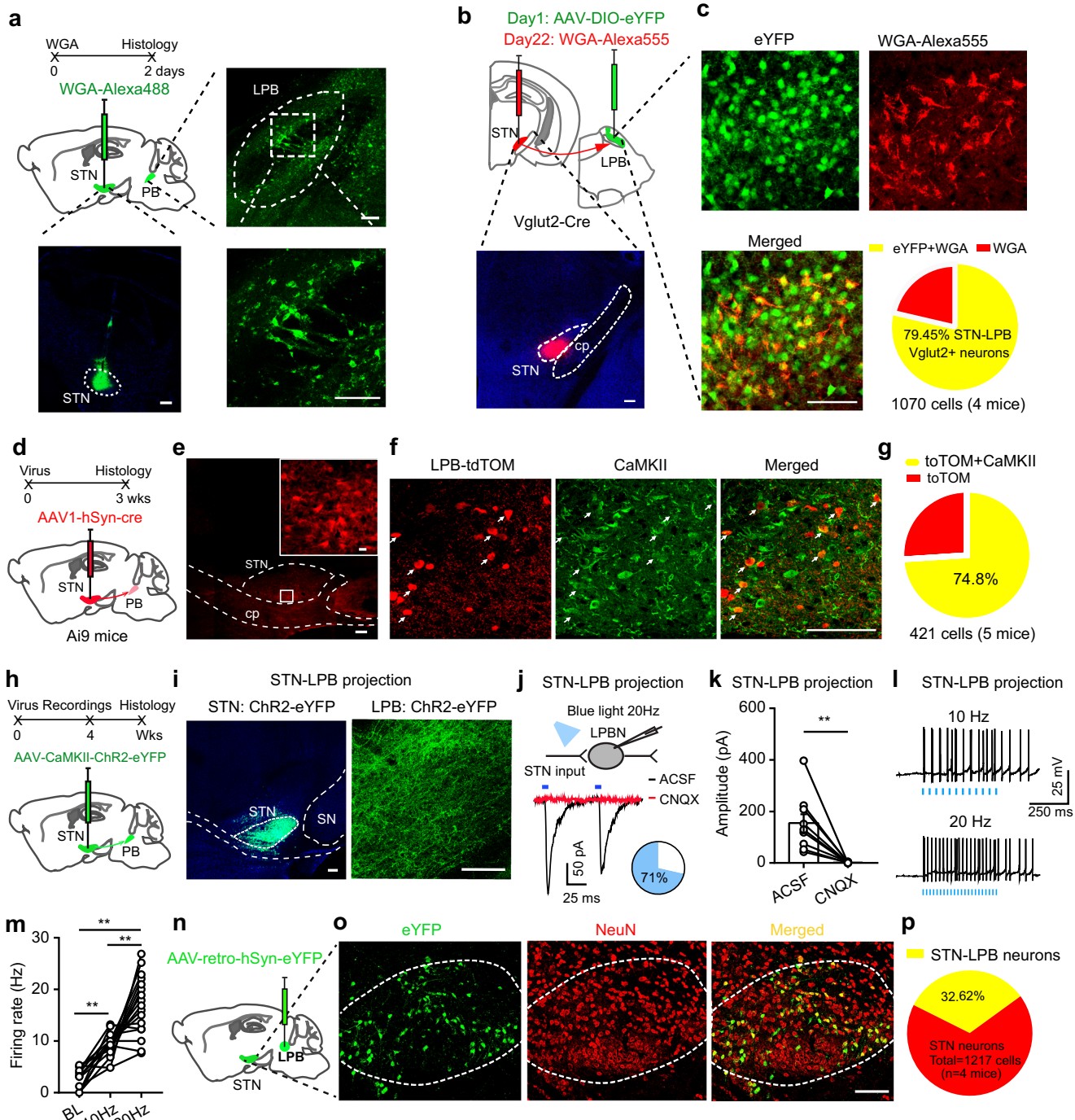

**Fig. 3 | STN^Glu neurons directly innervate LPB^Glu neurons. a** Anterograde labeling of lateral parabrachial nucleus (LPB) neurons from the STN. **b** Experimental diagram for labeling LPB glutamatergic neurons (upper) and an example image showing the WGA injection in the STN (lower). **c** Colocalization of WGA and Vglut2 in LPB neurons and the quantification pie chart. **d** Anterograde labeling of LPB neurons from the STN with scAAV1-hSyn-Cre. **e** Example images of an AAV1-Cre injection site in the STN. **f–g** Colocalization of tdTomato (tdTOM) and CaMKII in the LPB (**f**, representative images; **g**, quantification). **h, i** Viral vector was injected in the STN to label STN neuronal somata and projections to the LPB (**h**, schematic diagram; **i**, example images). **j** Whole-cell patch-clamp recordings were performed to record postsynaptic currents in LPB neurons evoked by paired pulses of 5 ms blue light stimulation (50 ms interval, 2 mW). The photo-EPSCs were blocked by 20 μM CNQX (trace in red). The pie chart indicates the percentage of responding neurons (blue). **k** Quantification of photo-EPSC amplitude in the absence and presence of CNOX ($t = 4.16$, $P = 0.003$, $n = 3$ mice (9 neurons) per group). **l, m** Corresponding firing response to 10 and 20 Hz blue light stimulation (**m**, $F_{(1.35, 21.53)} = 59.23$, $P < 0.0001$, $n = 3$ mice (17 neurons). **n** Experimental diagram of retrograde labeling of STN neurons from the LPB. **o, p** Example images (**o**) and quantification (**p**) of STN–LPB-projection neurons. Neurons were counted in 10 sections from 4 mice. **$P < 0.01$; Two-tailed paired $t$ test for (**k**). one-way ANOVA with Tukey's post-hoc analysis for (**m**). Data are presented as mean ± SEM. scale bars: 100 μm, except for that in the inset of panel (**e**) (10 μm).

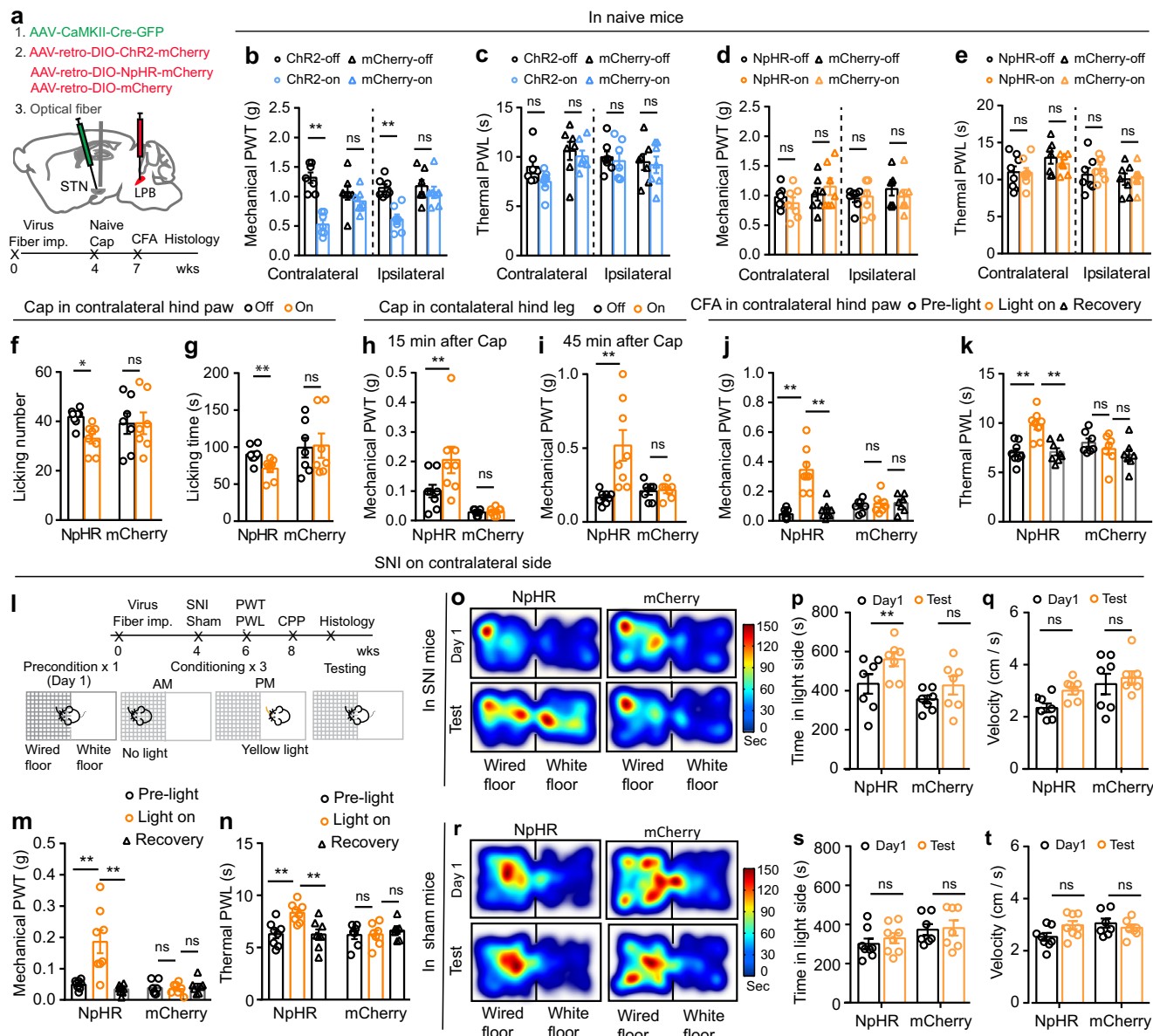

**Fig. 4 | STN–LPB neurons modulate pain-like behaviors. a** Experimental diagram for optogenetic manipulation of STN–LPB neurons. Timeline for panel (**b–k**). **b–e** Effect of activation or inhibition of STN–LPB neurons on pain thresholds in naive mice. **b** $F_{(1, 24)} = 35.4$, $P < 0.0001$. **c** $F_{(1, 24)} = 2.98$, $P = 0.1$. **d** $F_{(1, 24)} = 0.077$, $P = 0.78$. **e** $F_{(1, 25)} = 0.029$, $P = 0.87$; $n \geq 7$ per group. **f–g** Effect of silencing of STN–LPB neurons on the frequency (**f**, $F_{(1, 13)} = 7.57$, $P = 0.01$) and duration (**g**, $F_{(1, 13)} = 5.46$, $P = 0.03$) of nocifensive behavior induced by hind paw injection of capsaicin. $n \geq 7$ per group. **h, i** Effect of silencing of STN–LPB neurons on PWT 15 min (**h**, $F_{(1, 13)} = 5.53$, $P = 0.035$) and 45 min (**i**, $F_{(1,13)} = 7.70$, $P = 0.016$) after hind leg injection of capsaicin. $n \geq 7$ per group. **j, k** Effect of silencing of STN–LPB

neurons on PWT (**j**, $F_{(2, 26)} = 30.27$, $P < 0.0001$) and PWL (**k**, $F_{(2, 26)} = 14.09$, $P = 0.0003$) in CFA mice. $n \geq 7$ per group. **l** Timeline and experimental diagram for panels (**m–t**). **m, n** Effect of silencing of STN–LPB neurons on PWT (**m**, $F_{(2, 26)} = 14.25$, $P < 0.0001$) and PWL (**n**, $F_{(2, 26)} = 14.04$, $P = 0.0001$) 2 weeks after SNI. $n \geq 7$ per group. **o–t** Representative heat maps (**o**, **r**), time spent (**p**, **s**) and velocity (**q**, **t**) in the yellow-light-paired chamber in the pre-test and test session for SNI and sham ($n \geq 7$ per group) mice. **p** $F_{(1, 12)} = 14.72$, $P = 0.0024$. **q** $F_{(1, 12)} = 4.28$, $P = 0.061$. **s** $F_{(1, 13)} = 0.65$, $P = 0.44$. **t** $F_{(1, 13)} = 1.03$, $P = 0.33$. *$P < 0.05$. **$P < 0.01$; Two-way ANOVA with Tukey's post-hoc analysis for (**b–k**, **m**, **n**, **p**, **q**, **s** and **t**). Data are presented as mean ± SEM.

pain may reflect cellular and/or synaptic changes within the STN–LPB pathway. We thus performed experiments to explore this possibility. We used AAV-CaMKII-ChR2-eYFP to label STN neurons (Fig. 8a) and recorded photo-EPSCs in LPB neurons in brain slices while activating STN terminals by delivering blue light when the membrane voltage was held at −50 mV (Fig. 8b). The peak amplitude of photo-EPSCs was significantly increased in SNI mice relative to sham control mice (Fig. 8b, c), suggesting that SNI enhances the STN–LPB projection. The firing rate of LPB neurons was also increased in SNI mice relative to sham control mice (Fig. 8d, e), whereas the resting membrane potential and membrane input resistance did not differ between these two

groups of neurons (Supplementary Table 1). To investigate whether the enhancement of the STN–LPB projection observed in neuropathic pain involves presynaptic or postsynaptic mechanisms, or both, we measured the paired-pulse ratio (PPR) between the second and first EPSCs evoked by a pair of blue light pulses (5 ms, 2 mW, 50 ms interval) (Fig. 8f). We found that the PPR was dramatically reduced in LPB neurons in SNI mice relative to sham control mice (Fig. 8g). These data suggest that SNI enhances the STN–LPB projection at least partially through presynaptic mechanisms. Thus, the hyperactive STN neurons in SNI mice (Fig. 1) may provide a drive for enhancement of the STN–LPB projection.

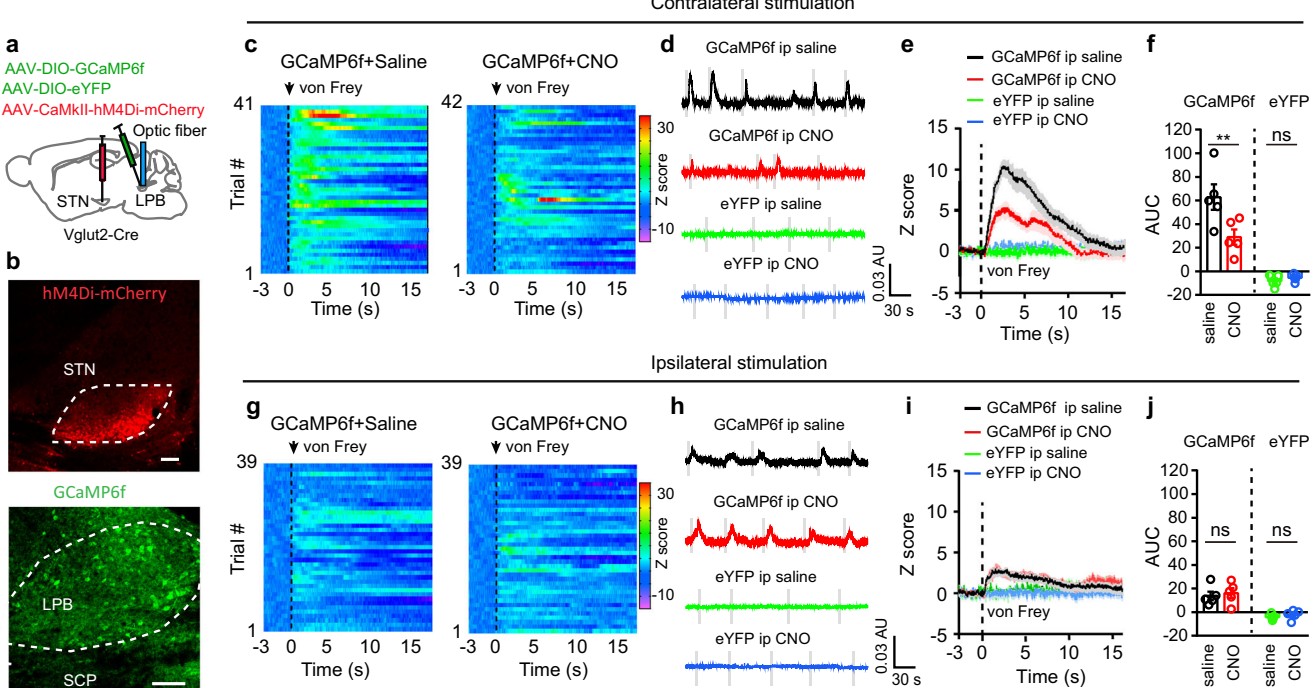

**Fig. 5 | Chemogenetic silencing of STN neurons reduces the response of LPB[Glu] neurons to peripheral mechanical stimulation. a** Experimental approach for GCaMP6 signal recordings of LPB[Glu] neurons with chemogenetic inhibition of STN neurons. **b** Example images of hM4Di and GCaMP6f expression in the injection sites. **c–j** Heat maps (**c, g**), example traces (**d, h**), average traces (**e, i**), and quantification (**f, j**, AUC in panels **e** and **i**) of GCaMP6f signal in the LPB of mice receiving

von Frey stimulation (4 g) of hind paws with i.p. injection of saline or CNO (3 mg/kg). **f** $F_{(1,8)} = 7.69$, $p = 0.02$, $n = 5$ mice. **j** $F_{(1,8)} = 1.06$, $p = 0.33$; $n = 5$ mice. CNO or the same volume of saline was applied 45 min prior to GCaMP6f signal recording. *$P < 0.05$, **$P < 0.01$; Two-way ANOVA with Tukey's post-hoc analysis for (**f** and **j**). Data are presented as mean ± SEM. AU in panels (**d**) and (**h**) stands for arbitrary unit of fluorescence intensity. Scale bars: 100 μm.

## Neuropathic pain induces hypoactivity in SNr[GABA] neurons

Based on our recent study demonstrating the SNr–STN projection confers an analgesic effect in several pain conditions[37], we hypothesized that compromised inhibitory synaptic inputs to STN neurons may be associated with their hyperactivity in various pain states. Indeed, we found a reduced frequency but not amplitude of miniature inhibitory postsynaptic currents (mIPSCs) in STN neurons, compared with those in sham mice (Fig. 8h–j). We then combined optogenetics and brain slice patch-clamp recordings to address whether major or potential GABAergic input nuclei of the STN are modified in the neuropathic pain state, contributing to the hyperactivity of STN neurons. First, we injected AAV-DIO-ChR2-eYFP into the GPe, SNr, or PPN of Vgat-Cre mice to label local GABAergic neurons (Fig. 8k, l; Supplementary Fig. 7a, b, g, h). Optogenetic activation of GABAergic terminals from the SNr, GPe, or PPN in the STN elicited IPSCs (photo-IPSCs) in STN neurons in brain slices and these photo-IPSCs were blocked by bicuculline (BIC, 10 μM), a GABA_A receptor antagonist (Fig. 8m; Supplementary Fig. 7c, i). Mice expressing ChR2-eYFP in GABAergic neurons in the SNr (SNr[GABA]), GPe (GPe[GABA]), or PPN (PPN[GABA]) were subjected to contralateral SNI or sham surgery. We found that the spontaneous and evoked firing rates were significantly reduced in SNr[GABA] neurons, but not in GPe[GABA] or PPN[GABA] neurons between 1 and 2 weeks after SNI surgery relative to sham mice (Fig. 8n–p; Supplementary Fig. 7d–f, j–l). There was no difference in resting membrane potential and membrane input resistance in SNr[GABA] neurons, GPe[GABA] or PPN[GABA] neurons between sham and SNI mice (Supplementary Table 2). These data suggest that hypoactivity of SNr[GABA] neurons in SNI mice may contribute to the attenuated inhibitory input to STN neurons.

## Connectivity of the SNr[GABA]-STN[Glu]-LPB[Glu] pathway

We next characterized the anatomical and functional connectivity of the SNr[GABA]-STN[Glu]-LPB[Glu] pathway. First, we took advantage of a

modified rabies virus (RV)-mediated Cre-dependent retrograde monosynaptic tracing system[52]. The Cre-dependent helper viral vectors (AAV-EF1a-DIO-TVA-GFP and AAV-EF1a-DIO-RVG) were injected into the STN of Vglut2-Cre mice. Three weeks later, RV-EnvA-ΔG-DsRed was injected into the LPB (Fig. 9a, b). Seven days later, we sacrificed the mice and observed DsRed-labeled neurons in the SNr (Fig. 9c). To further examine the functional synaptic connectivity of the SNr–STN–LPB pathway, we injected AAV-DIO-ChR2-eYFP into the SNr of Vgat-Cre mice to label SNr[GABA] neurons and AAV-retro-hSyn-mCherry into the LPB to label STN–LPB neurons (Fig. 9d). We observed mCherry-labeled neurons and eYFP-labeled fibers in the STN (Fig. 9e). As shown by whole-cell patch-clamp recordings in the STN slices, brief light stimulation of ChR2-containing SNr[GABA] terminals reliably elicited photo-IPSCs (125.2 ± 18.02 pA) on 68% of mCherry-labeled STN neurons (17/25 cells), and the photo-IPSCs were blocked by BIC (Fig. 9f–h). Taken together, these data reveal a functional circuit, the SNr[GABA]-STN[Glu]-LPB[Glu] pathway, which connects the basal ganglia with a pain pathway.

## The SNr[GABA]-STN[Glu]-LPB[Glu] pathway in pain-like hypersensitivity

To ascertain how this pathway is involved in modulation of pain signals, we selectively transfected hM3Dq or mCherry in STN-projecting SNr neurons (SNr–STN neurons) by injecting retrograde AAV-retro-hSyn-Cre-eGFP in the STN and AAV-DIO-hM3Dq-mCherry or AAV-DIO-mCherry in the SNr. These mice were then injected with AAV-CaMKII-GCaMP6f in the LPB for in vivo fiber photometry recordings (Fig. 9i, j, k; Supplementary Fig. 8a, b, k, l). Increase of the GCaMP6f signal was recorded in the LPB following contralateral hind paw stimulation, and this increase was significantly attenuated when SNr–STN neurons were activated with CNO (Fig. 9l–o; Supplementary Fig. 8c–f; Supplementary Fig. 11d–f). However, CNO did not change the GCaMP6f signal following ipsilateral hind paw stimulation (Fig. 9p–s; Supplementary

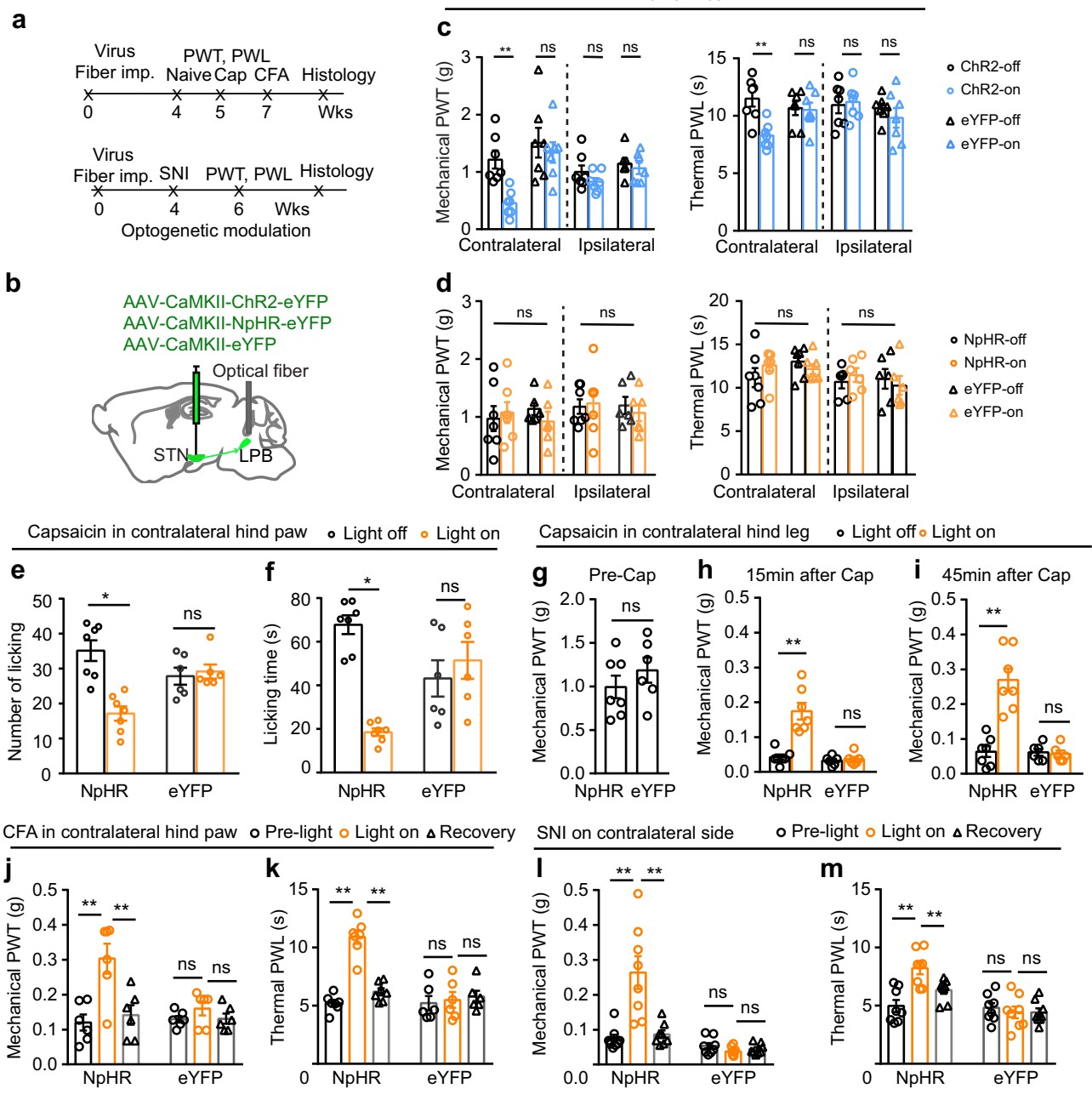

**Fig. 6 | The STN–LPB projection modulates pain-like behaviors. a, b** Timeline and diagram for optogenetic manipulation of the STN–LPB projection. **c** Effect of activation of the STN–LPB projection on pain thresholds in ChR2-expressing mice. PWT: $F_{(1, 24)} = 12.02$, $P = 0.002$; PWL: $F_{(1, 12)} = 10.46$, $P = 0.0035$; $n = 7$ per group. **d** Effect of silencing of the STN–LPB projection on pain thresholds in NpHR-expressing mice. PWT: $F_{(1, 22)} = 0.4$, $P = 0.53$; PWL: $F_{(1, 22)} = 0.14$, $P = 0.71$; $n \geq 6$ per group. **e, f** Effect of silencing of the STN–LPB projection on the frequency ($F_{(1, 11)} = 19.16$, $P = 0.0011$) and duration ($F_{(1, 11)} = 44.53$, $P < 0.0001$) of licking/flinching behavior induced by capsaicin hind paw injection. $n = 7$ in NpHR, $n = 6$ in eYFP.

**g–i** Effect of silencing of the STN–LPB projection on PWT and PWL 15 min and 45 min after injection of capsaicin into hind legs. $n \geq 6$ per group. **g** Baseline PWT, $t = 0.99$, $P = 0.34$; **h** $F_{(1, 11)} = 19.02$, $P = 0.0011$; (**i**) $F_{(1, 11)} = 61.07$, $P < 0.0001$. **j, k** Effect of silencing of the STN–LPB projection on PWT ($F_{(2, 22)} = 8.13$, $P = 0.0025$) and PWL ($F_{(2, 22)} = 9.96$, $P = 0.001$) 24 h after CFA injection in hind paws. $n \geq 6$ per group. **i–m** Effect of silencing of the STN–LPB projection on PWT ($F_{(2, 28)} = 14.44$, $P < 0.0001$) and PWL ($F_{(2, 28)} = 17.24$, $P < 0.0001$) 2 weeks after SNI surgery. $n = 8$ per group. *$P < 0.05$. **$P < 0.01$; Two-way ANOVA with Tukey's post-hoc analysis for (**c–f**) and (**h–m**); Two-tailed paired $t$ test for (**g**). Data are presented as mean ± SEM.

Fig. 8g–j). These effects are similar to the effect that the STN–LPB projection exerts on LPB neurons.

Next, we examine whether this pathway modulates pain threshold. STN neurons within this pathway were selectively ablated by dual injection of AAV-retro-CaMKII-Cre-mCherry in the LPB and AAV-FLEX-taCasp3-TEVP[53] in the STN (Fig. 10a). This leads to expression of taCasp3 only in STN–LPB neurons and subsequent selective ablation of these neurons. In the same mice, AAV-GAD67-ChR2-eGFP or AAV-

GAD67-eYFP was then transfected in SNr^GABA neurons and an optical fiber was implanted in the STN (Fig. 10a, b; Supplementary Fig. 9a, e, f, g). This procedure greatly decreased the number of STN neurons (Fig. 10c, d); however, baseline mechanical and thermal thresholds and locomotion were not affected (Fig. 10e–h; Supplementary Fig. 9b–d). Then the mice were subjected to contralateral lower hind leg capsaicin injection or SNI to establish acute inflammatory and neuropathic pain models, respectively (Fig. 10a). The pain thresholds were measured according to the

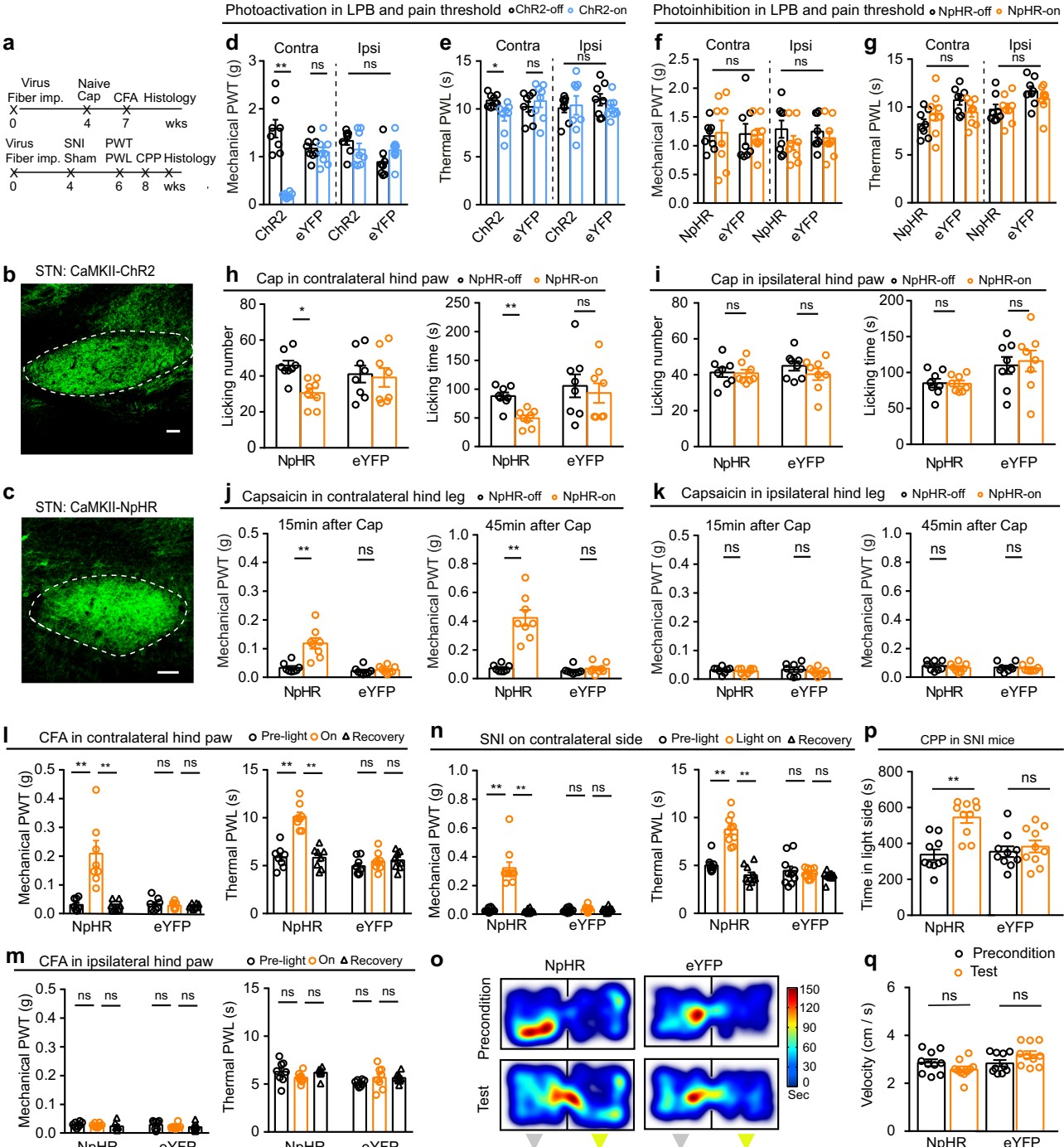

**Fig. 7 | The STN–LPB projection modulates pain-like behaviors in female mice.**
**a** Timeline of experimental setup. **b, c** Example images of ChR2 (**b**) and NpHR (**c**) expressions in the STN. **d–g** Effect of activation or inhibition of the STN–LPB projection on pain thresholds ($n = 8$ per group). **d** $F_{(1, 28)} = 20.8$, $P < 0.0001$; **e** $F_{(1, 28)} = 1.19$, $P = 0.28$; **f** $F_{(1, 28)} = 0.57$, $P = 0.46$; **g** $F_{(1, 28)} = 0.51$, $P = 0.48$. **h, i** Effect of silencing of the STN-LPB projection on nocifensive behavior induced by capsaicin injection into hind paws ($n = 8$ per group). **h** Left, $F_{(1, 14)} = 5.35$, $P = 0.03$; Right, $F_{(1, 14)} = 7.88$, $P = 0.01$. **i** Left, $F_{(1, 14)} = 0.88$, $P = 0.36$; Right, $F_{(1, 14)} = 0.18$, $P = 0.68$. **j, k** Effect of silencing of the STN-LPB projection on PWT induced by Cap injection into hind legs ($n = 8$ per group). **j** Left, $F_{(1, 14)} = 23.6$, $P = 0.0003$; Right, $F_{(1, 14)} = 29.3$, $P < 0.0001$. **k** Left: $F_{(1, 14)} = 0.15$, $P = 0.70$; Right,

$F_{(1, 14)} = 0.11$, $P = 0.75$. **l, m** Effect of silencing of the STN–LPB projection on pain threshold in CFA mice ($n = 8$ per group). **l** Left, $F_{(2, 28)} = 17.47$, $P < 0.0001$; Right, $F_{(2, 28)} = 20.2$, $P < 0.0001$. **m** Left, $F_{(2, 28)} = 0.17$, $P = 0.84$; Right, $F_{(2, 28)} = 2.37$, $P = 0.11$. **n** Effect of silencing of the STN–LPB projection on pain thresholds 2 weeks after SNI. Left, $F_{(2, 36)} = 50.73$, $P < 0.0001$; Right, $F_{(2, 36)} = 43.08$, $P < 0.0001$; $n = 10$ per group. **o–q** Silencing of the STN–LPB projection induced CPP ($n = 10$ per group). **p** $F_{(1, 18)} = 9.69$, $P = 0.006$; **q** $F_{(1, 18)} = 0.15$, $P = 0.7$. Gray and yellow triangles in panel (**o**) represent no light and yellow light presented in the chamber during the conditioning session, respectively. **P < 0.01; Two-way ANOVA with Tukey's post-hoc analysis for (**d–n, p**, and **q**). Data are presented as mean ± SEM. Scale bars: 100 μm.

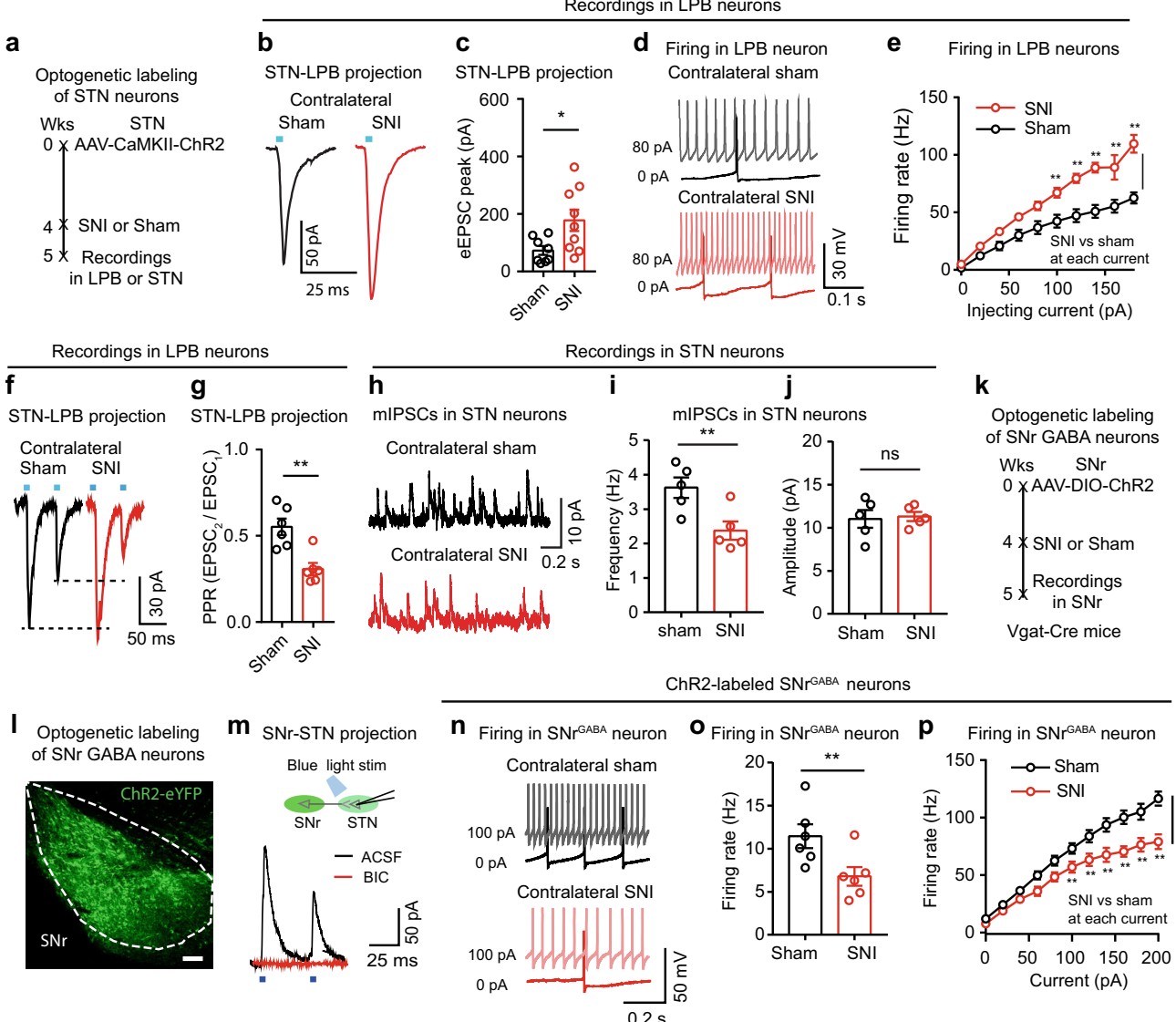

**Fig. 8 | Modification of STN^Glu–LPB^Glu projection in neuropathic pain. a** Timeline of experiments for panels (**b**–**j**). **b**, **c** Example traces (**b**) and peak amplitude (**c**) of blue light (5 ms, 2 mW)-evoked photo-EPSCs in LPB neurons from sham and SNI mice. **c** $t = 2.49$, $P = 0.025$, $n = 9$ per group. **d**, **e** Example traces (**d**) and evoked firing (**e**) in LPB neurons in week 2 after sham or SNI. Group: $F_{(1, 175)} = 100.1$, $P < 0.0001$. **f**–**g** Example traces (**f**) and paired-pulse ratio (PPR) (**g**) of blue light (5 ms, 2 mW, 50 ms interval)-evoked photo-EPSCs recorded in LPB neurons in slices from sham and SNI mice. **g** $t = 4.15$, $P = 0.002$, $n = 6$ per group. **h**–**j** Example traces (**h**) and quantification (**i**, **j**) of mIPSCs in STN neurons in week 2 after sham or SNI ($n = 5$ per group). Frequency (**i**): $t = 3.1$, $P = 0.011$; Amplitude (**j**): $t = 0.24$, $P = 0.82$. **k** Timeline

of experiments for recording from SNr^GABA neurons in sham and SNI mice. **l** Example image showing ChR2-eYFP expression in Vgat(+) neurons in the SNr. **m** Whole-cell patch clamp recording of blue light-evoked photo-currents in STN neurons (top). Photo-currents in STN neurons (bottom) were blocked by 10 μM bicuculline (BIC), a GABA_A receptor blocker. **n** Firing recorded from SNr^GABA neurons in week 2 after sham or SNI on the contralateral side. **o**, **p** Summary of spontaneous (**o**) and evoked firing (**p**) recorded from SNr^GABA neurons from sham ($n = 6$, 13 cells) and SNI ($n = 6$, 16 cells) mice. **o** $t = 2.33$, $P = 0.027$. **p** Group: $F_{(1, 234)} = 94.93$, $P < 0.0001$. *$P < 0.05$, **$P < 0.01$; Two-tailed $t$ test for (**c**, **g**, **i**, **j** and **o**); Two-way ANOVA with Tukey's post-hoc analysis for (**e** and **p**). Data are presented as mean ± SEM. Scale bar: 100 μm.

timeline as shown in Fig. 10a, in the absence and presence of photo-stimulation of the SNr–STN projection. Ablation of STN neurons prevented the pain-relieving effect of activation of the SNr–STN–LPB pathway 15–30 min and 45–60 min after contralateral capsaicin injection (Fig. 10i, j, k) and 2 weeks after SNI (Fig. 10l, m).

Furthermore, optogenetic stimulation of the SNr–STN projection in SNI mice resulted in place preference to the chamber paired with blue light illumination, but not in SNI mice in which STN–LPB neurons had been lesioned via the taCasp3 system (Fig. 10n–p). Control experiments confirmed that expression of eYFP in the SNr and light delivery in the STN did not affect pain thresholds in capsaicin or SNI mice and did not elicit CPP or aversion in SNI mice (Fig. 10i–q).

These data suggest that SNr^GABA neurons control the regulation of pain-like hypersensitivity via the STN^Glu-LPB^Glu pathway.

Moreover, we examined the modulatory effect of the SNr–STN–LPB pathway on physiological pain thresholds and pain-like hypersensitivity in pathological pain states in female mice (Supplementary Fig. 10a–c). Consistent with those in male mice (Fig. 10; Supplementary Fig. 9), optogenetic stimulation of the SNr–STN–LPB pathway did not change mechanical and thermal pain threshold, performance in the adhesive tape removal test, and motor function in the open field test in naïve female mice (Supplementary Fig. 10d–h, q–u, v), but mitigated mechanical hypersensitivity in capsaicin female mice (Supplementary Fig. 10j, k). Female mice with SNI also showed attenuated mechanical and thermal hypersensitivity and developed CPP

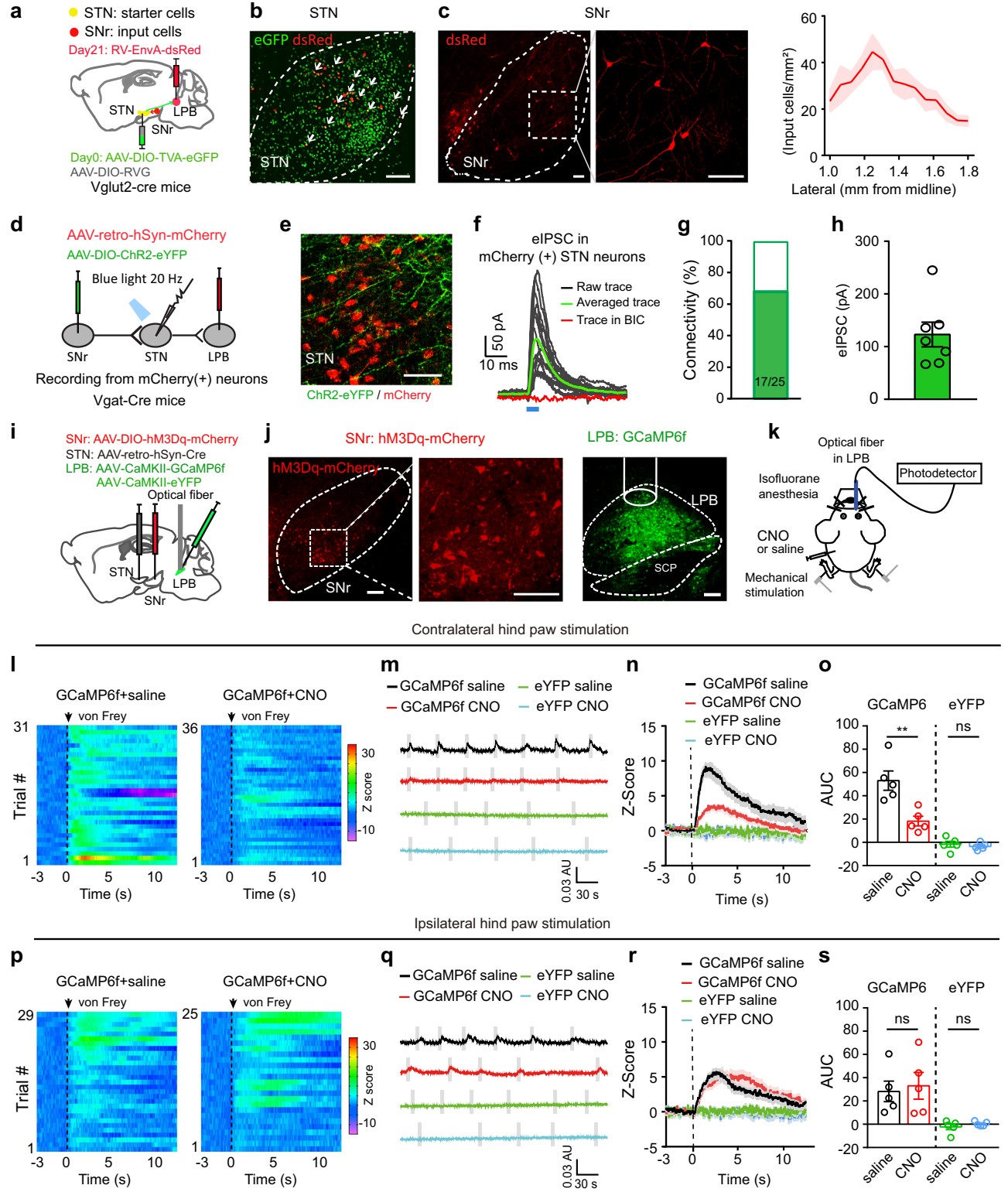

upon optogenetic stimulation of this pathway (Supplementary Fig. 10l–p).

These data suggest that the SNr^GABA-STN^Glu-LPB^Glu pathway also regulates pain-like hypersensitivity in female mice.

## Discussion

The contribution of the STN to pain modulation has been studied in the context of Parkinson's disease, in which STN activity is pathologically enhanced in humans and rodents[20,23,26,54–56]. In this study, we

demonstrate that a subset of neurons in the STN not only participate in the processing of sensory pain signals but also modulate pain-like hypersensitivity in inflammatory and neuropathic pain models (Figs. 1, 2, 4). We also provide several lines of morphological and functional evidence showing that STN^Glu neurons project directly to LPB^Glu neurons (Figs. 3, 6, 7), which are known to be involved in pain processing, and that the projection is further controlled by SNr^GABA neurons (Fig. 9). Furthermore, peripheral nerve injury leads to strengthening of the STN^Glu-LPB^Glu projection as a result of

**Fig. 9 | Connectivity of the SNr^GABA–STN^Glu–LPB^Glu pathway. a** Experimental diagram of Cre-dependent monosynaptic retrograde rabies virus tracing. **b** Example images of viral expression in the STN. Starter cells (white arrows) co-expressing AAV-DIO-TVA-eGFP (green) and RV-EnvA-ΔG-DsRed (red). **c** Example images and quantification ($n = 5$ mice) of DsRed-labeled neurons in the SNr traced from LPB-projecting STN neurons. **d** Experimental diagram of patch clamp recording of STN–LPB neurons with blue light stimulation of ChR2-labeled SNr terminals in the STN. **e** mCherry-labeled STN neurons were surrounded by ChR2-labeled SNr terminals. **f** Blue light-evoked currents in mCherry(+) STN neurons were blocked by BIC ($n = 7$ mice). **g** Bar graph showing the percentage of responding neurons recorded in (**e**). **h** Amplitude of photo-IPSCs in mCherry(+) STN neurons ($n = 7$ mice). **i** Experimental diagram of virus injections. **j** Example images of hM3Dq and GCaMP6f expression at the injection sites. **k** Experimental diagram of GCaMP6f signal recordings in response to bilateral von Frey stimulation (4 g) of the hind paws and chemogenetic activation of STN-projecting SNr neurons with CNO (i.p., 3 mg/kg). CNO was applied 45 min prior to GCaMP6f signal recording. **l–s** Heat maps (**l, p**), example traces (**m, q**), average traces (**n, r**), and quantification (**o, s**, AUC in panels **n, r**) of GCaMP6f response in the LPB of mice receiving von Frey stimulation of hind paws after i.p. administration of saline or CNO. **o** $F_{(1,8)} = 13.36$, $p = 0.007$. **s** $F_{(1,8)} = 1.69$, $p = 0.023$; $n = 5$ per group. **$P < 0.01$; Two-way ANOVA with Tukey's post-hoc analysis for (**o**) and (**s**). Data are presented as mean ± SEM. AU in panels (**m**) and (**q**) stands for arbitrary unit of fluorescence intensity. Scale bars: 100 μm.

hypoactivity in the upstream SNr^GABA neurons (Fig. 8). Reversing the dysfunction in this pathway effectively mitigates pain-like hypersensitivity (Fig. 10). The present study advances our understanding of how the basal ganglia modulate pain perception and are involved in acute and persistent pain states.

Our fiber photometry data show that activity in STN neurons was enhanced in response to both mechanical and thermal stimulation of the bilateral hind paws (Fig. 2). This result is in line with a previous report showing that individual neurons in the STN respond to noxious stimuli with increased firing rate[26]. Note that both studies measured STN responses in anesthetized rodents. In this situation, the activation of STN neurons may represent sensation of pain but not motor responses to stimuli. Furthermore, neurons in the STN were hyperactive in acute and persistent pain states, as evidenced by increased c-Fos expression and the mechanical stimulation-induced $Ca^{2+}$ elevation (Figs. 1, 2). We further provide multiple lines of evidence at both the circuit and behavioral levels to show that STN neurons modulate pain processing at least partially through their projection to the LPB (Figs. 4, 6, 7), a brain-stem structure that contributes to both hyperalgesia and persistent pain[34,57,58]. We demonstrated that the STN sends a direct glutamatergic projection to the LPB that is sufficient to modulate LPB glutamatergic neurons; moreover, inactivation of the STN neurons attenuated activation of LPB^Glu neurons in response to mechanical stimulation (Fig. 5). We found that peripheral inflammation or nerve injury led to enhanced excitability in the STN (Fig. 1) and subsequent enhancement of glutamatergic output from the STN to the LPB (Fig. 8). At the behavioral level, activation of the STN–LPB projection resulted in pain-like hypersensitivity in naïve mice (Figs. 4, 6, 7), whereas optogenetic inhibition of the STN–LPB projection attenuated acute and persistent pain-like hypersensitivity (Figs. 4, 6, 7). These results are consistent with the hypothesis that inactivation of STN glutamatergic terminals attenuates the activity of LPB^Glu neurons to cause analgesia. We observed an inconsistent phenomenon that modulation of STN–LPB neurons regulated mechanical but not thermal threshold on both sides (Fig. 4), but modulation of the STN–LPB projection regulated both mechanical and thermal thresholds on the contralateral side (Figs. 6, 7). We postulate that modulation of STN–LPB neurons may affect other pathways besides the STN–LPB projection (Supplementary Fig. 3i–k), probably leading to different effects on pain modalities on either side. Interestingly, the mice with selective ablation of STN–LPB neurons still developed mechanical and/or thermal hypersensitivity after capsaicin injection or SNI surgery (Fig. 10; Supplementary Fig. 10). These data hint that the STN–LPB pathway does not mediate the initiation of acute and persistent pain conditions. Consistently, we fully acknowledge that the LPB also receives projections from other brain nuclei, such as the lateral hypothalamus and the central nucleus of the amygdala, and that these connections contribute to the processing and modulation of pain signals[13,59]. It is possible that different pathways may be integrated and engaged in different types of acute and persistent pain states.

Combining tract tracing, electrophysiological recording, and in vivo fiber photometry experiments, we demonstrated that the STN–LPB pathway is controlled by SNr^GABA neurons: activation of the SNr^GABA-STN^Glu projection suppressed activation of LPB^Glu neurons induced by von Frey filament stimulation and increased pain thresholds in pain states (Figs. 9, 10; Supplementary Fig. 10). Ablation of the STN–LPB projection prevented the analgesic effects of optogenetic modulation of the SNr^GABA-STN projection (Fig. 10; Supplementary Fig. 10). In SNI mice, we observed hypoactivity in SNr neurons, attenuation of the SNr^GABA-STN projection, enhancement of the STN^Glu-LPB projection, and hyperactivity in STN and LPB neurons (Fig. 8). Together with our neuromodulation data, these results suggest that modification of the SNr^GABA-STN^Glu-LPB^Glu pathway may be a pathophysiological basis of hyperalgesia in SNI mice. That is, hypoactivity of SNr^GABA neurons boosts STN glutamatergic inputs to LPB^Glu neurons, increasing activation of LPB^Glu neurons and facilitating pain signaling.

Previous studies have reported that SNr neurons respond diversely to various pain modalities and that using pharmacological or electrical neuromodulation to inhibit the STN or stimulate the SNr produces analgesic effects[60]. A recent study demonstrated that SNr neurons receiving direct nociceptive signals from excitatory neurons in the LPB are essential for pain perception[61]. Similarly, we previously reported that SNr neurons innervated by the STN partially mediate hyperalgesia in parkinsonian mice[20]. Contrary to these studies[20,61], but similar to another study[37], we observed that stimulation of SNr neurons led to analgesic effects. These results suggest that SNr neurons are diverse in terms of their involvement in pain perception, depending on which neural pathway they belong to. That is, those innervated by the LPB and STN may facilitate the transmission of pain signals, whereas those innervating the STN suppress pain perception. Indeed, two subsets of SNr neurons that respectively innervate STN neurons and are innervated by STN neurons are largely separated[37]. Taking the previous studies together with our current finding, we propose that SNr^GABA neurons receive pain-related information from the LPB and, in turn, regulate the downstream STN–LPB pathway and pain responses. However, in pain states, SNr^GABA neurons are hypoactive and lose the ability to initiate feedback inhibition of the STN–LPB pathway, resulting in hyperalgesia.

Spontaneous ongoing pain is one major symptom presented in patients with neuropathic pain. The CPP paradigm has been used to evaluate whether a method mitigates spontaneous pain[48]. We observed that stimulation of the SNr–STN projection and inhibition of the STN–LPB projection established a CPP in SNI-NpHR mice but inhibition of STN–LPB neurons did not induce CPP in sham-NpHR mice (Figs. 4, 10; Supplementary Fig. 10), suggesting that these manipulations alleviate spontaneous pain, but do not cause rewarding effect (Fig. 4). These data indicate that, in addition to modulating pain thresholds, the SNr^GABA-STN^Glu-LPB^Glu pathway regulates spontaneous pain in neuropathic pain condition.

Accumulating evidence implicates the basal ganglia in pain processing[8,16–20,26,33,60,62]. Most studies have focused on the dorsal and ventral striatum. Several studies have demonstrated that both the STN and SNr receive projections from excitatory neurons in the LPB and are involved in nociception[26,61]. In addition to our previous study showing

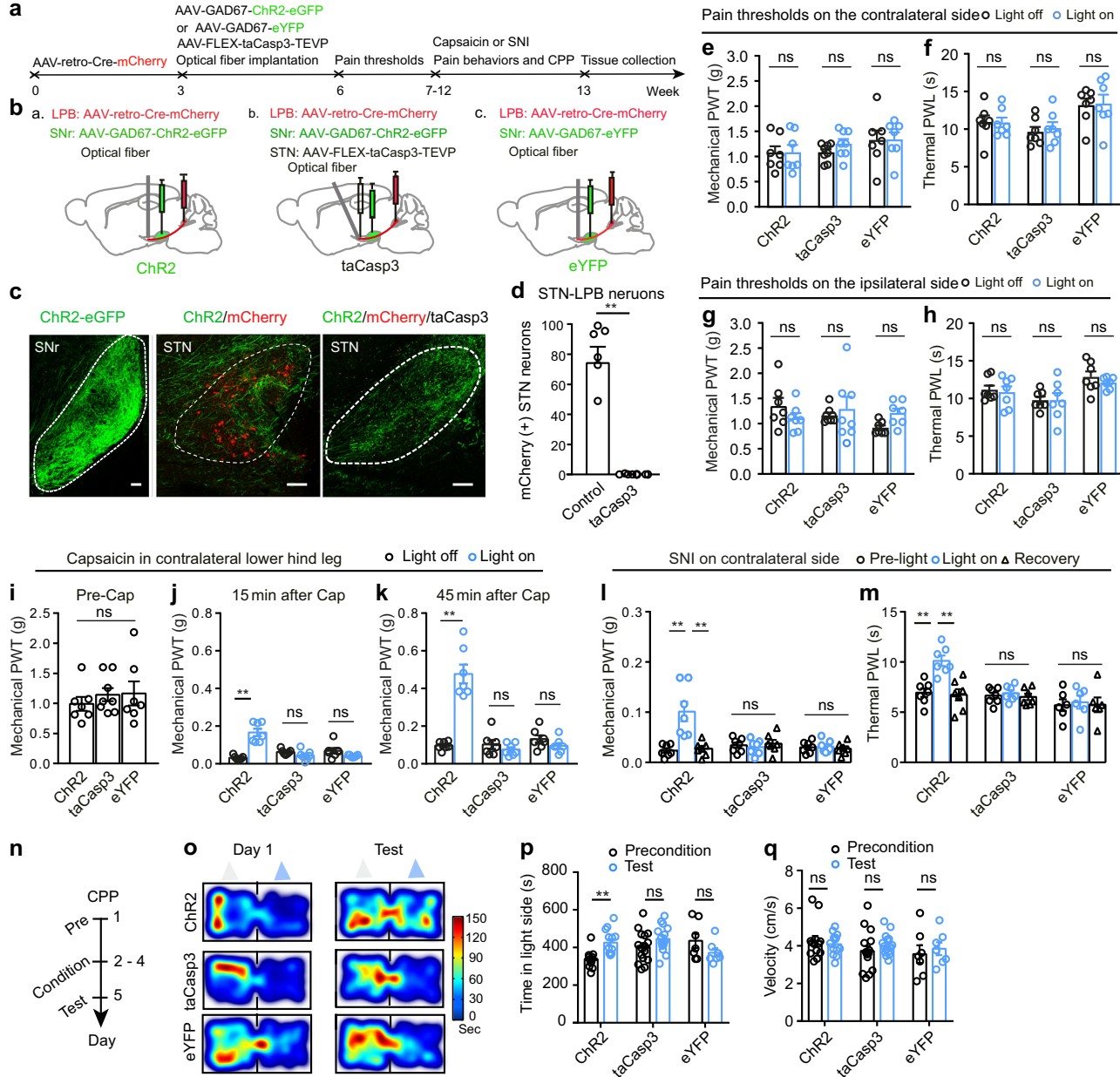

**Fig. 10 | The SNr^GABA-STN^Glu-LPB^Glu pathway in pain-like behaviors. a, b** Timeline and diagram of experimental setup. **c** Example images of ChR2-expression in the SNr (left), mCherry-labeled STN−LPB neurons surrounded by ChR2-labeled SNr^GABA terminals (middle), and ChR2-labeled SNr^GABA terminal in the STN with ablation of STN−LPB neurons (right). **d** Quantification of the loss of STN−LPB neurons ($t = 7.28$, $P = 0.0003$, $n = 6$ per group). **e−h** Effect of disruption of the SNr−STN−LPB pathway on PWT and PWL on both hind paws. **e** $F_{(1, 19)} = 0.56$, $P = 0.58$. **f** $F_{(1, 19)} = 0.087$, $P = 0.92$. **g** $F_{(2, 19)} = 2.44$, $P = 0.11$. **h** $F_{(2, 19)} = 0.18$, $P = 0.84$. $n \geq 7$ per group. **i** Baseline PWT in the three groups ($F_{(2, 19)} = 0.42$, $P = 0.66$; $n \geq 7$ per group). **j−m** Effect of ablation of STN−LPB neurons on the pain-relieving effects of activation of the SNr−STN projection in mice with hind leg injection of capsaicin or SNI surgery. **j** $F_{(2, 19)} = 12.22$, $P = 0.0004$. **k** $F_{(2, 19)} = 27.38$, $P < 0.0001$. **i** ($F_{(4, 40)} = 14.99$, $P < 0.0001$. **m** $F_{(4, 40)} = 4.87$, $P = 0.0029$. $n = 7$ per group. **n** Timeline of the CPP experiment. **o−q** Representative heat maps (**o**), time spent (**p**) and velocity (**q**) in the blue-light-paired chamber in day 1 and test session in SNI mice. **p** $F_{(2, 33)} = 6.42$, $P = 0.004$. **q** $F_{(2, 33)} = 1.0$, $P = 0.38$. $n \geq 7$ per group. Gray and blue triangles in panel (**o**) represent no light and blue light presented in the chamber during the conditioning session, respectively. **$P < 0.01$; Two-tailed $t$ test for (**d**); One-way ANOVA with Tukey's post-hoc analysis for (**i**); two-way ANOVA with Tukey's post-hoc analysis for (**e−h**, **j−m**, **p** and **q**. Data are presented as mean ± SEM. Scale bars: 100 µm.

that the STN regulates the transmission of pain signals and is implicated in central sensitization in parkinsonian hyperalgesia[20], the present study demonstrates the existence of the SNr^GABA-STN^Glu-LPB^Glu pathway and a modulatory role of this pathway in multiple pain modalities and several pain states in both male and female mice (Supplementary Fig. 12). Our results suggest that disruption of this pathway may be a potential therapeutic strategy for the treatment of acute and persistent pain. Concerning the circuit mechanisms of pain

modulation, this study presents evidence for a direct connection through which the basal ganglia modulate a pain pathway.

## Methods

### Study approval

The care and use of animals and the experimental protocols (No. L202008A086) used in this study were approved by the Institutional Animal Care and Use Committee and the Office of Laboratory Animal

Resources of Xuzhou Medical University under the Regulations for the Administration of Affairs Concerning Experimental Animals (1988) in China.

## Animals

Vglut2-IRES-Cre, Vgat-IRES-Cre, and Ai9 transgenic mice were purchased from the Jackson Laboratory. Heterozygous transgenic mice were bred with wild-type C57BL/6J (WT) mice in animal facility at Xuzhou Medical University and the heterozygous offspring were used for viral vector injections. The mice were group-housed (no more than 5 per cage) at stable temperature of $23 \pm 2 \, ^{\circ}\text{C}$ on a 12-h light/dark cycle with ad libitum access to water and food. Both male and female mice at least 8-week old were used in the experiments. Efforts were made to minimize animal suffering and to reduce the number of animals used.

## Viral vectors

AAV-CaMKII-GCaMP6f, AAV-CaMKII-eYFP, AAV-CaMKII-NpHR-eYFP, AAV-DIO-eYFP, AAV-CaMKII-ChR2-eYFP, AAV-CaMKII-Cre-eGFP, AAV-DIO-TA-TVA-eGFP, AAV-DIO-RVG, RV-EnvA-ΔG-dsRed, AAV-EF1α-FLEX-taCasp3-TEVP and self-complementary (sc) AAV1-hSyn-Cre were purchased from Brain VTA (Wuhan, China). AAV-retro-DIO-ChR2-mCherry, AAV-EF1α-DIO-ChR2-eYFP, AAV-retro-DIO-NpHR-mCherry, AAV-EF1α-DIO-hM3Dq-mCherry, AAV-EF1α-DIO-mCherry, AAV-CaM-KII-hM4Di-mCherry, AAV-GAD67-ChR2-eGFP, AAV-GAD67-eYFP, AAV-retro-hSyn-Cre, AAV-retro-EF1α-DIO-mCherry, and AAV-retro-EF1α-Cre-mCherry were purchased from OBIO (Shanghai, China). The viral titers are $(2-9) \times 10^{12}$ (viral genomes per ml) for AAV and $2 \times 10^{8}$ (viral genomes per ml) for RV.

## Stereotaxic surgeries and injection

Mice were deeply anesthetized with intraperitoneal injection of sodium pentobarbital, placed on a heating pad, and stabilized on a stereotaxic apparatus (RWD Life Science Co., Ltd, Shenzhen, China). Small holes were drilled in the skull above brain regions of interest. Injections (200 nL of virus per site at 50 nL/min) were made using an automatic microinjection pump (KD Scientific, Holliston, MA, USA) or (World Precision Instruments, Sarasota, FL, USA).

The coordinates (relative to the Bregma) for viral injection were as follows: STN (AP, −1.85 mm; ML, 1.48 mm; DV, −4.75 mm), LPB (AP, −5.2 mm, ML, 1.2 mm, DV, −3.5 mm), SNr (AP, −3.1 mm; ML, 1.5 mm; DV, −4.7 mm), GPe (AP, −0.35 mm; ML, 2.15 mm; DV, −3.65 mm), and PPN (AP, −4.3 mm; ML, 1.3 mm; DV, −3.5 mm). Optical fiber implants (200 μm in diameter, NA 0.37) (Inper, Hangzhou, China) were placed 200 μm above (for optogenetic manipulation) or in (for fiber photometry recordings) the injection site and were fixed to the skull with dental cement. Mice with virus injections and implants were allowed to recover for at least 3 weeks before electrophysiological recordings and morphological assays. Viral expression and the position of fiber implants in each mouse were confirmed histologically after the termination of the experiments. We only included mice with viral expression confined to the SNr, STN, LPB, GPe or PPN and optical fibers in right places for either optogenetic modulation and fiber photometry recordings (Supplementary Figs. 1a, b, 2g, j, 3r, s, 5c, k, 6a, b, 8k, l, 9e–g, 10d, and 10v).

To anterogradely trace the STN–LPB projection, WGA (0.1%, w/v, 100 nl, conjugated with Alexa488 or Alexa555, Thermo Fisher Scientific, Waltham, MA, USA)[20] was injected into the STN and 48 h recovery was allowed for sufficient anterograde transportation; scAAV1-hSyn-Cre (100 nl)[63] was injected into the STN of Ai9 mice.

Rabies-virus-mediated retrograde tracing[52] was used to examine the SNr–STN–LPB connection. The mixture of AAV2/9-DIO-TVA-EGFP and AAV2/9-DIO-RVG (1:2 ratio) was injected into the STN of Vglut2-Cre mice. Three weeks later, RV-EnvA-DsRed was injected into the LPB. Mice were sacrificed 7 d later.

Ketaprofen (20 mg/L) in drinking water had been provided ad libitum for 3-day post-operative pain relief.

## Fiber photometry

A fiber photometry instrument (ThinkerTech, Nanjing, China)[64,65] was used to monitor the GCaMP6f signals in STN neurons or LPB neurons. We adjusted the instrument by setting the excitation light to 50 μW and the gain to a level that gave a background signal of 3 units measured when the end of the input optical cable was in the dark. After the input optical cable was connected to the optical implant in the mouse brain with a ceramic sleeve, the instrument read the total light signals. The difference between the total signal and the background signal was used as the baseline GCaMP6f signal. To evaluate responses of STN and LPB neurons to von Frey filament or heating stimuli, we designated 3 s GCaMP6f signal prior to the response as the baseline value ($F_0$); the peak responses in the GCaMP6 signal traces were quantified as [peak signal $(F) - F_0]/F_0$. To summarize the response, we calculated the mean and standard deviation (SD) of 3 s GCaMP6 signal prior to the response and used these parameters to calculate the Z-score ((F–Mean)/SD) for each point in the GCaMP6f trace, thus transforming GCaMP6f signal into Z-score trace. We then measured the area under the curve (AUC) of the Z-score plot to quantify the response of STN neurons and LPB neurons to sensory stimuli.

## Optogenetic manipulation

For ChR2-mediated optogenetic stimulation, 473-nm laser pulses (5 ms, 20 Hz, 4 mW) were delivered. For NpHR-mediated optogenetic inhibition, a 3 mW 589-nm laser was kept on continuously for 1–2 min. All optogenetic manipulations were performed unilaterally in the right hemisphere. Therefore, the contralateral side refers to the left side of the body and the ipsilateral side refers to the right side.

## von frey filament test

Individual mice were acclimatized for at least 1 h in a test compartment on a wide gauge wire mesh supported by an elevated platform. von Frey filaments with fiber force between 0.1–4 g were used to measure mechanical paw withdrawal threshold (PWT) of both hind paws. The 50% threshold was determined by the up–down method[20,37,66].

## Thermal nociception threshold

Individual mice were acclimatized for at least 1 h in a test compartment on a glass surface. Thermal paw withdrawal latencies (PWL) in both hind paws were measured with a plantar anesthesia tester (Boerni, Tianjin, China)[20,37,67].

## Capsaicin-induced nocifensive behaviors and secondary mechanical hyperalgesia

Capsaicin-induced inflammatory pain models were established according to previous reports[37,46]. Capsaicin (0.01%, 20 μL in 10% DMSO/saline) was injected in the plantar area of the hind paw. Spontaneous pain behaviors (licking/flinching the hind paw) were recorded over 15 min. These pain behaviors were essentially absent after 15 min. In separate experiments, capsaicin was injected subcutaneously in the lower hind leg to induce secondary mechanical hyperalgesia in the plantar area of the hind paw.

## Complete Freund's adjuvant (CFA) injection

CFA (20 μL, Sigma-Aldrich) was injected in the plantar area of the hind paw to induce inflammatory pain and hyperalgesia[37].

## Spared nerve injury (SNI)

Spared nerve injury (SNI), a neuropathic pain model, was established according to previous reports[37,68]. Mice were deeply anesthetized with an intraperitoneal injection of sodium pentobarbital, the fur in the operation area from the knee to the hip was shaved, and the skin was sterilized with 75% alcohol. A longitudinal incision was made in the

shaved area and a blunt dissection was performed in the biceps femoris muscle to expose the sciatic nerve and its branches (sural, common peroneal, and tibial nerves). The common peroneal and tibial nerves were tightly ligated with two nylon sutures separated by 3 mm and 2 mm nerves in between were cut and removed. The mice were allowed to recover on a heating pad. Mice not subjected to nerve ligation and nerve severing were used as sham controls. For pain threshold measurements, the *von Frey* filaments and heat beam were targeted to the skin area innervated by the sural nerve.

## Locomotion
Locomotor activity of individual mice in a round open field arena (30 cm in diameter and 40 cm in height) was recorded with a video camera controlled by Ethovision XT 14.0 software[20,64,69]. For analysis, a circle (15 cm in diameter) in the center of the arena was defined as the central zone.

## Conditioned place preference (CPP) test
The CPP was performed in a custom-made two-chamber box (length × width × height: $40 \times 20 \times 30 \, cm^3$): the right chamber had black-and-white vertical stripes on the walls and a smooth floor and the left chamber had black-and-white horizontal stripes on the walls and a mesh floor. The CPP test was performed according to a previous study[69] with some modifications.

Day 1 was the preconditioning test (pre-test) day in which mice were given free access to the two chambers and we recorded the time that the mice spent in each chamber. On days 2–4, the mice were restricted to one chamber (counterbalanced across all mice) and received either constant illumination (stimulation) or no stimulation for 20 min. Approximately 4 h later, the mice were restricted to the other chamber and received the opposite treatment (no stimulation or stimulation) for 20 min. On day 5 (test day), the mice were allowed to freely explore the chambers for 20 min and the time spent in each chamber was recorded. On the pre-test and test days, the animal's movement was video-tracked and analyzed online or offline with EthoVision XT video tracking software. We calculated the time spent in the light-paired side on the pre-test and test days. Mice were not used if they spent more than 75% of the total time in one chamber on the pre-test day.

## Adhesive tape removal test
Adhesive tape removal test was performed as described in a previous study[20]. Mice were habituated for at least 30 min in a transparent chamber (15 cm long, 10 cm wide, and 15 cm high). A round adhesive tape (8 mm in diameter) was attached to the plantar surface of the left hind paw. Time latencies for mice to contact the tape (sense time) and to remove the tape (removal time) were recorded. The mice performed this task once per day for 5 consecutive days for training before they were enrolled in optogenetic modulation.

## Brain slice electrophysiology
These experiments were carried out according to previously reported method[20,37,64]. Parasagittal slices (300 μm) containing STN, LPB, SNr, GPe, and PPN were cut with a vibratome (VT-1200S, Leica Microsystems) in ice-cold modified sucrose-based artificial cerebral spinal fluid (sACSF), saturated with 95% $O_2$ / 5% $CO_2$ (carbogen) containing (in mM) 85 NaCl, 75 sucrose, 2.5 KCl, 1.25 $NaH_2PO_4$, 4.0 $MgCl_2$, 0.5 $CaCl_2$, 24 $NaHCO_3$, and 25 glucose. Slices were allowed to recover in sACSF at 32 °C for 75 min, and then in carbogenated normal ACSF containing (mM) 125 NaCl, 2.5 KCl, 1.2 $NaH_2PO_4$, 1.2 $MgCl_2$, 2.4 $CaCl_2$, 26 $NaHCO_3$, and 11 glucose at room temperature for at least 30 min before use.

Neurons in brain slices were visualized under an upright microscope (FN-1, Nikon, Tokyo, Japan) equipped with a CCD-camera (Flash 4.0 LTE, Hamamatsu, Hamamatsu city, Japan). Whole-cell patch-clamp signals were recorded from neurons with a MultiClamp 700B amplifier,

a Digidata 1550B analog-to-digital converter, and pClamp 10.7 software (Molecular Devices, San Jose, CA, USA). The patch electrode had a resistance of 4 – 6 MΩ when filled with a low-chloride intrapipette solution containing (in mM) 135 K gluconate, 0.2 EGTA, 0.5 $CaCl_2$, 10 HEPES, 2 Mg-ATP, and 0.1 GTP, pH: 7.2; osmolarity: around 300 mOsm. After formation of whole-cell recordings, the neurons with a holding current bigger than −50 pA and with a resting membrane potential more depolarized than −40 mV were not included in the analysis.

Evoked EPSCs (eIPSCs) or mIPSCs in the presence of tetrodotoxin (TTX, 1 μM) were recorded at −50 mV. To verify glutamatergic or GABAergic connections, APV (50 μM) plus CNQX (20 μM) or bicuculline (10 μM) was bath-applied. Firings following current injections (1 s, 20–200 pA steps with a 20 pA increment and a 30 s inter-sweep interval) were recorded in the current-clamp mode.

To obtain light-evoked responses, blue light (460 nm, 2 mW) or yellow light (560 nm, 2 mW) was delivered through an optical fiber (200 μm, NA 0.37) connected to a PlexBright LED light source (Plexon Inc., Hong Kong, China).

## Histology
Mice were sacrificed in a $CO_2$ chamber and then subjected to cardiac perfusion with phosphate-buffered saline (PBS), followed by 4% paraformaldehyde (PFA) in PBS. Mouse brains were removed and post-fixed in PFA for 4–6 h at 4 °C. Brain samples were cut into 80 μm sections with a Leica VT-1200s vibratome and mounted onto glass slides. For immunostaining, brain sections were incubated in a blocking buffer containing 5% donkey serum and 0.1% Triton for 90 min at room temperature. Then the sections were incubated with primary antibody diluted in blocking buffer for 24 h at 4 °C (rabbit anti-c-Fos IgG, 1:2000, Cell Signaling Technology; Guinea pig anti-NeuN IgG, 1:1000, Millipore; Mouse anti-CaMKII IgG, 1:1000, Cell Signaling Technology). After washing three times (10 min each) in PBS, the sections were incubated with secondary antibodies (Alexa 555-conjugated donkey anti-rabbit IgG, Alexa 555-conjugated donkey anti-guinea pig IgG) for 90 min at room temperature. The sections were washed three times (10 min each) in PBS, dried in the dark, and then cover-slipped in mounting medium (Meilunbio, Dalian, China).

The sections were imaged with a confocal microscope (LSM 880, Zeiss) and the images were processed with ImageJ (NIH)[70].

## Timeline of behavioral, histological, and electrophysiological tests in SNI mice
Sensory hypersensitivities in SNI mice appear within 3 days after nerve injury, from then on, pain thresholds in these mice remain at a very low level (Fig. 1). We assessed the role of the SNr–STN–LPB pathway in SNI-induced neuropathic pain at 2 time points: mid-term (>7 days) and long-term (4 weeks). During mid-term, pain thresholds remain constant over time, therefore, pain threshold measurement, motor function assay, c-Fos staining, and patch-clamp recordings in SNI or sham mice were performed > 7 days, mostly between 2–3 weeks after surgery. As anxiety- and depression-like behaviors arise after several weeks after SNI surgery, CPP was performed during long-term.

## Chemicals
Complete Freund's adjuvant (CFA) and Capsaisin (Cap) were purchased from Sigma-Aldrich. DL-2-Amino-5-phosphonovaleric acid lithium salt (APV), bicuculline methobromide (BIC), and 6-Cyano-7-nitro-quinoxaline-2, 3-dione disodium salt hydrate (CNQX) were purchased from Tocris.

## Statistics
GraphPad Prism 7.0 was used for statistical analyses. Clampfit 10.7 (Molecular Devices) was used for analysis of electrophysiological and GCaMP6 data. Figures were prepared with Adobe Illustrator CS6. All data are expressed as mean ± SEM. Two-tailed paired or unpaired *t*

tests were used for comparison of parameters between two groups if data were normally distributed. ANOVAs followed by Tukey's post-hoc analysis were used for multiple comparisons. If the equal-variance assumptions were not valid, statistical significance was evaluated with the Mann–Whitney test or ANOVA rank tests. The mean and SEM, n (the number of animals), the specific statistical test, and F and P values are reported in the figure legends. A value of $P < 0.05$ was considered statistically significant. The number of mice used in each experiment was calculated in a priori power analysis (StatMate 2.0), the power of each experiment was set to 0.8.

## Reproducibility

Experiments were repeated independently with similar results at least three times. Representative images from experiments were repeated independently: Fig. 2a (5 times), 3a–c (4 times), 3e (5 times), 3i (>10 times), 5b (5 times), 7b (8 times), 7c (8 times), 8i (5 times), 9b–c (6 times), 9e (7 times), 9j (5 times), 10c (6 times).

## Reporting summary

Further information on research design is available in the Nature Portfolio Reporting Summary linked to this article.

## Data availability

The data for supporting the findings of this study are available from the corresponding authors upon request. Source data are provided with this paper.

## Code availability

There is no custom-written code or software used in this study.

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

## Acknowledgements

This work was supported by the National Key R&D Program of China-the Sci Tech Innovation 2030 Major Project (2021ZD0203100, J.L.C.), the National Natural Science Foundation of China (81971038 (C.Z.), 82171235 (C.Z.), 81870891 (C.X.), 82071231 (C.X.), 82271293 (C.X.), 82293641 (J.L.C.), 82130033 (J.L.C.)), the Fund for Jiangsu Province Specially-Appointed Professor (C.X., C.Z.), the Natural Science Foundation of Jiangsu Province (BK20211349) (C.Z.), and Leadership Program in Xuzhou Medical University (JBGS202203) (C.X., C.Z.). T.J. and J.C. acknowledge the Postgraduate Innovation Program in Jiangsu Province (KYCX22_2936, KYCX22_2952).

## Author contributions

C.Z., C.X., and J.L.C. designed and supervised this research. C.Z. and C.X. collected and analyzed electrophysiological data. T.J., J.C., Y.W., and X.Z. performed mouse survival surgeries, morphological experiments, behavioral tests, and managed mouse colony. T.J., J.C., Y.W., and C.Z. analyzed behavioral and imaging data. C.Z., C.X., T.J. and J.L.C. wrote the manuscript. All authors read and approved the manuscript.

## Competing interests

The authors declare no competing interests.
