## [Peer Review File · Nature Communications]

A nigro–subthalamo–parabrachial circuit modulates pain-like behaviorsREVIEWER COMMENTS

Reviewer #1 (Remarks to the Author):

Pain is a common source of distress and lack optimal treatments. Recent studies have revealed new nociceptive circuits outside the classical pain pathways able to modulate nociceptive processing. The general idea of this manuscript is to test whether these circuits could represent potential therapeutic targets for effective treatment of chronic pain. The clinical impact of this work is crucial as clinical management of pain remains difficult, especially for chronic pain.

The new targeted network involves two main structures of the basal ganglia, the subthalamic nucleus (STN) and the substantia nigra pars reticulata (SNr) in relation to a primary nociceptive structure from the brainstem, the lateral parabrachial nucleus (LBN). This network has been mainly studied in the context of pain symptoms in Parkinson's disease. The present work focused on the anatomical interconnectivity and involvement of this group of structure in the modulation of chronic pain.

This manuscript is dense and includes a huge amount of anatomical, behavioral and electrophysiological studies, with additional data in supplementary materials... The main striking discovery of this work is (i) the description of two new pathways, one between the STN and the PBN and the other between the SNr and the STN, and (ii) the demonstration of the role of both pathways to modulate the circuits electrophysiologically, and acute and chronic pain behaviorally.

The results can be subdivided into two main parts

Part 1: STN – LPB pathways

Experience 1: Role of STN in acute and chronic pain: STN neurons are hyperactive in acute and chronic pain with an increase of c-fos expression in acute and chronic pain models and present an exacerbated evoked firing rate in brain slices recordings.

Experience 2: In vivo investigation of STN dynamic changes in response to pain stimulation using AAV-CaMKII-GCaMP6f injection in the STN and GCaMP6f signal recordings. The results confirm results from experience 1 and suggest the involvement of STN Glutamatergic neurons.

Experience 3: Dissection of the circuit underlying the involvement of the STN in pain-like behavior with (i) Characterization of STN-PBN pathway using anterograde (Alexa488-conjugated WGA) and retrograde (not indicated in the paper, you have to look in the supplementary material) tracing; (ii) characterization of LPB neuronal types innervated by STN. 80 % are glutamatergic. The results confirm the existence of a new pathway between the STN and LPB.

Experience 4: Characterization of STN-LPB projection functionality using (i) AAV-CaMKII-ChR2-eYFP injected into STN, optogenetic stimulation of STN infected neurons, and slice recording of LPB. LPB firing rate was increased by photo-stimulation of the STN and (ii) c-fos immunohistochemistry showing an increased number of c-fos positive neurons in the LPB. The results confirm STN ability to modulate LPB neuronal activity.

Experience 5: Exploration of STN-LPB pathway contribution to pain processing using optogenetic modulation (excitatory and inhibitory) targeting specifically this pathway using retro-virus. The aim is to test the effect of specific STN-LPB pathway on pain response. The results support that stimulation of STN-LPB neurons induces mechanical hypersensitivity. Inhibition of STN-LPB pathway had no effect.

Experiment 6: Test whether inactivation of STN-LPB pathway can ameliorate pain-like hypersensitivity. In acute and chronic pain states, using same technique as before but in acute and chronic pain models. The results show that silencing STN-LPB pathway regulates nociceptive responses behaviorally for acute pain but also in chronic pain states that involve central sensitization.

Experiment 7: Test whether the STN-LPB pathway modulate the emotional component of pain using pain-relief-conditioned place preference behavioral apparatus. The results show that silencing STN-LPB activity also attenuated the emotional components of pain.

Experiment 8: To test whether the previous effects are due to a specific recruitment of STN-LPB pathway, without the contaminating involvement of axonal input to other nuclei. The author modulated STN terminals in the LPB using optogenetic and behavioral assessment. The author concluded that the activation of the STN-LPB projection is sufficient to reduce pain-threshold but play a limited role in maintenance of the normal pain threshold.

Experiment 9: Test the role of STN-PBN projection in modulation of acute and chronic pain.

From experiment 8 and 9: the authors concluded that STN-LPB projection control pain-like hypersensitivity through axonal inputs to the LPB rather than through input to other regions outside the LPB. These experiments are not clear enough to understand the conclusion of the authors.

Experiment 10: To test if the STN-LPB glutamatergic projection is involved in pain-like behavior. The results suggest that disrupting neuronal activity in the STN is sufficient to reduce nociceptive responses in LPB^{Glu} neurons.

Experiment 11: To test if the ability of STN to relieve pain may reflect cellular and/or synaptic changes within STN-LPB pathway since chronic pain has been associated with changes in the plasticity of the pain transmission pathways. They evaluated whether enhancement of the STN-LPB projection observed in neuropathic pain involves presynaptic or post-synaptic mechanisms using paired pulse ratio. The results show that the paired pulse ratio was reduced in LPB neurons in SNI mice compared to sham mice. The authors concluded of pre-synaptic mechanisms to enhance STN-LPB activity in a chronic pain model.

Part 2: The authors then looked for the origin of STN hyperactivity in acute and chronic pain.

Experiment 12: Test whether GABAergic neurons activity in the Pedunculo pontin nucleus (PPN), Globus pallidus external (GPE) and/or Substantia nigra pars reticulata (SNr) is modified in a neuropathic pain state. The authors found that SNr neurons, but not GPe or PPN neurons are hypoactive in SNI mice compared to sham mice, suggesting SNr attenuated inhibitory input to STN may represent a major contributor for STN hyperactivity.

Experiment 13: To characterize the anatomical and functional connectivity of the SNr^{GABA}, STN^{Glu} and LPB^{Glu} pathway. The results confirm that alteration of the SNr input to the STN modulates nociceptive responses in the LPB.

Experiment 14: to test whether SNr^{GABA} neurons modulate the nociceptive responses in LPB via the STN.

Experiment 15: To test whether the ablation of STN prevents the pain-relieving effect of activation of the SNr-STN-LPB pathway.

Experiment 16: To test whether the stimulation of SNr-STN pathway has an impact on the emotional component of pain in a CPP paradigm.

This summary was meant to illustrate the amount of data squeezed into a small format manuscript. There are two very important stories in this manuscript and the crucial discovery made by the authors are not properly highlighted in the present form.

My suggestion would be to split the two parts of the results into two different articles. This will allow to facilitate the reading of the manuscript and the results.

Part 1: A single manuscript on the STN-LPB pathway would help to clarify the aim and conclusion of the experiment 8 and 9 (see above). The authors should be careful to use the right terminology between nociception and pain. It would also be crucial to add a thorough histological analysis including for example all the injection sites of their viruses. The attempt to present all the data was done to some important methodological and histological disadvantage.

Part 2: This set of experiment needs clarification on several aspects: The authors have focused their experiments of STN firing pattern control toward an inhibitory hypothesis. However, this control may be more complex than with GABA inputs only. For example, it has been shown that cortico – subthalamic pathway provide STN firing rate (Magill et al., 2000; Bolam et al., 2002) while this is a glutamergic input. Excitatory input to the STN may also modulate STN GABAergic interneuron to modulate STNGlu neuronal activity. Looking for the origin of STN hyperactivity may require more experiments to complete the story.

There is a misunderstanding for several anatomical pathways described by the authors. The authors cited two articles to justify GPe, SNr and PPN GABA input to the STN.

- Benarroch EE, Neurology, 2008

- Yang et al. Nature Neuroscience, 2021.

While GPe – STN pathway is indeed well known, the connection between the PPN and the STN are reciprocal and involves Glutamatergic and Cholinergic inputs (Benarroch, 2008), not GABAergic. The connection between SNr and STN is not described in the previous papers and to my knowledge has never been characterized. The SNr-STN pathway is not well known as claimed by the authors. This is why the results from part 2 are really interesting as it would represent another reciprocal loop similar to those described between the striatum and the GPe, and between the GPe and STN. This is also why I would suggest to the authors to thoroughly analyze this new circuit anatomically and functionally, physiologically and in the context of pain control in a separate manuscript.

Finally, STN^{GLU} neurons are hyperactive and are well known to project to SNr^{GABA} neurons. This hyperactivity should, in theory, lead to a hyperactivity of SNr neurons. How can the authors explain SNr hypoactivity with this connectivity?

Reviewer #2 (Remarks to the Author):

In this manuscript titled "A nigro-subthalamo-parabrachial pathway modulates pain-like behaviors", the authors explored the involvement of substantia nigra pars reticulata (SNr)- subthalamic nucleus (STN)- lateral parabrachial nucleus (LPB) pathway in acute mice inflammation and neuropathic pain models. The authors proved that STNGlu neurons are activated after injury and can be activated by peripheral von Frey fiber stimulation (Figure 1), STNGlu neurons directly innervate LPBGlu neurons (Figure 2), optogenetically activation of STNGlu neurons can induce hypersensitivity to non-noxious von Frey stimulation in naïve rats while optogenetically inactivation of STNGlu neurons takes analgesic effects in mice with painful injury (Figure 3), optogenetically inactivation of terminals of STNGlu neurons in LPB also takes the similar analgesic effects (Figure 4), chemogenetically inhibition of STN reduced the response of LPBGlu neurons to mechanical stimulation by von Frey fibers in naïve rats (Figure 5). Then they found that there are increased activity of STNGlu and LPBGlu neurons, and decreased activity of SNrGABA neurons (Figure 6). They further prove the SNrGABA-STNGlu-LPBGlu pathway (Figure 7). Finally, they selectively ablated STN neurons, which prevented the pain-relieving effect of activation of the SNr-STN-LPB pathway (Figure 8). They concluded that this SNrGABA- STNGlu- LPBGlu is critical for the development of mechanical and thermal hypersensitivity in acute and 'chronic' pain states. This is an interesting and well-designed study, which provides valuable findings to this field. I have the followings suggestions/questions.

1. The authors proved the SNrGABA-STNGlu-LPBGlu pathway (Figure 1-7). Then, they selectively ablated STN neurons, which prevented the pain-relieving effect of activation of the SNr-STN-LPB pathway (Figure 8). If STNGlu neuronal activity is critical in the development of mechanical and thermal hypersensitivity after painful nerve injury, is there any explanation on that those mice with ablation of STNGlu neurons still developed hypersensitivity to von Frey and heat after nerve injury (Figure 8), which indicates that activity of STNGlu neurons is not critical in the pain development and and contradictory to their main point? It seems that this pathway is only one of many regulatory pathways in the brain in pain condition.

2. Regarding the concept of acute pain and chronic pain, 7 days after SNI should be still in acute phase of neuropathic pain, even though it is debatable on the timing of transition from acute to chronic pain. According to a study (PMID: 34244365), there is dynamic change of neuronal activity after nerve injury. Please correct this misconception about chronic pain in the whole manuscript.

1 day after Complete Freund's adjuvant (CFA) paw injection induced pain cannot also be classified as chronic pain, as they claimed in FigureS1-B.

3. Regarding the concept of CPP test, CPP test is not an indicator of emotional component of neuropathic pain. Any effective analgesic treatment can induce CPP, even it only targets peripherally. So, CPP positive is only an indicator of reduced spontaneous pain, not emotional component of pain. Please correct it thoroughly.

4. It is not clear what the purpose of experiments of GCaMP6f without injury (Fig. 1I-P) is, even though it was claimed to test "the dynamic change of the activity of STN neurons in response to pain stimulation". Which part of the data represent the dynamic change of the activity? Mechanical stimulation with von Frey fibers is non-noxious. It is irrelevant to pain or nociception. They concluded that "these results suggest that STNGlu neurons encode nociception", which is not true as activated neurons after nerve injury doesn't mean they encode nociception. Their ablation data (Figure 8) also confirmed this

that STNGlu neurons are not critical in encode painful sensation.

5. Details on electrophysiological recordings are needed. Are those recordings on EPSC and IPSC recorded with GABAAR blockers or NMDAR and AMPAR blockers, respectively? What is the holding potential? What is reason that spontaneous IPSC (Fig. 6G-I), not miniature IPSC, was recorded to explore the pre- vs post- synaptic mechanisms? On the recording of firing in STN (Figure S1D-G) and LPB neurons (Fig. 6C, D), what is duration of the current injection? What is the normal resting membrane potential of these neurons? What is the standard to exclude recordings as it seems that many neurons recorded have abnormal resting membrane potentials, which is an indicator of bad recordings with too much leaking and contribute to the results dramatically especially in current clamp whole-cell recording?

6. SNr, STN, and LPB innervate each other in both directions. SNr receive innervations from STN and LPB, and send innervations to them. As the authors mentioned in the discussion "we propose that SNrGABA neurons receive pain related information from the LPB", shouldn't increased LPB neuronal activity after painful nerve injury will activate SNrGABA neurons, thus reduce STN activity and then reduce LPB activity? SNr neurons also receive innervations from STN. If STN neurons and LPB neurons are all activated after painful nerve injury, SNr neurons should be also activated, not deactivated, to provide inhibitory feedback regulation.

7. Please provide reference for this sentence "Contrary to this, we observed that stimulation of SNr neurons led to analgesic effects" in page 16, line 406.

Reviewer #3 (Remarks to the Author):

The manuscript describes a hitherto uncharacterized pathway that may regulate pain perception, from the subthalamic nucleus (STN) to the parabrachial nucleus (PB). This complements the previous characterized reciprocal pathway from PB to STN (which the authors, surprisingly, do not explicitly discuss). The authors apply an impressive array of behavioral, electrophysiological and functional imaging approaches to test the hypothesis that the STN->PB pathway is causally related to both acute and chronic pain. The findings are novel and of significance. However, a number of concerns reduce the rigor of the findings and their conclusion.

- That only male mice were used in a study of pain is inexcusable. For at least two decades the pain field has emphasized the stark sex differences in pain processing, and the clinically significant differences in signs and symptoms and in therapeutic approaches to pain. Studying only males should no longer be accepted.

- This (line 319) is a head scratcher: Ablating STN neurons "greatly decreased the number of neurons in the STN (Fig. 8C and 8D); however, baseline nociceptive sensitivity and locomotion were not affected (Fig. S7B – S7H)." Does this finding not disprove the overarching hypothesis of this study?

- In all experiments comparing data from sham and treated (e.g. SNI, CFA) animals, data (e.g. electrophysiological metrics) should be averaged from each animal, and

animals (as opposed to neurons) should be used as the counting objects for statistical analyses. The same holds also for analyzing cFos data, where animals, and not sections, should be counted. This applies, for example, to Figs. 1b,d,f; S1e,f,g; S3; S4; S6' 6, and many more.

- There is no mention of an a-priori power analysis, or any explicit consideration of statistical power.
- The authors appear to acknowledge that manipulating STN is likely to affect motor behaviors, including reflexive measures of nociception and non-reflexive pain behaviors. Relying only on locomotion, assessed in open field, is likely not sufficient to allay this concern.
- Preclinical models of neuropathic pain, such as SNI used here, do not consider one week post-surgery as the "chronic pain" period. The 7 to 10 days post-SNI period used to assess behavior and electrophysiology in some experiments needs to be justified. The authors should also justify why in some experiments they focused on 7 days post-SNI, and in others they used a 2 weeks post-SNI period.
- The hypothesis that STN activity is increased in persistent pain states is tested only using cFos labeling (Fig. 1). Why did the authors apply their GCaMP imaging approach to study only acute pain, and not also persistent pain states?
- Related to that, micrographs showing GCaMP expression in STN would be informative.
- Page 6, line 140: Provide a more descriptive detail of the finding, beyond stating "affect the activity of LPB neurons".
- Quantification in Fig. 2 is lacking or unclear. How many cells do the percentages represent? How many animals were assayed?
- The finding that mechanical, but not thermal responses were affected by modulating STN activity appears contrary to previous findings, including from the authors' lab. This should be discussed.
- Fig. 3(l-m): Please clarify what "recovery" refers to. For example, how long after optical activation?
- Figure 3(o-r) It appears that only SNI animals were used. The absence of sham animals should be justified. Equally importantly, the data in P and R panels should be compared directly using a 2-way ANOVA, and not treated as different statistical tests. This applies also to the data in Figure S4.
- The absence of sham controls for the data in Fig. 4 should be justified.
- The absence of CNO-only controls (without hM4Di) for the data in Figs. 5,7 should be justified.
- A graphical representation of experimental timelines for each experiment would be useful.

Dear Reviewers

We feel extremely thankful for your helpful critiques to improve our manuscript. To respond to your comments, we made great efforts to conduct more experiments, organize the image data, and correct flaws in figures and texts. Below, we included our point by point responses to the critiques. We divided the rebuttal letter into three parts for each reviewer and started each part with “Reviewer 1, 2, or 3”. We made your original comments italicized and began our responses with “Response:”. In revision of the manuscript, we kept the original structure of the manuscript to facilitate review process and attached new panels in the original figures or supplementary figures. For description of these new data, please see highlights in main text, supplementary information, and figure legends. We hope that we properly solved your concerns and you may feel that by incorporating changes and new data you suggested, we made substantial improvement in the manuscript.

Thank you again for the time and efforts you have put in this manuscript.

Sincerely yours

Chunyi Zhou

REVIEWER 1

Pain is a common source of distress and lack optimal treatments. Recent studies have revealed new nociceptive circuits outside the classical pain pathways able to modulate nociceptive processing. The general idea of this manuscript is to test whether these circuits could represent potential therapeutic targets for effective treatment of chronic pain. The clinical impact of this work is crucial as clinical management of pain remains difficult, especially for chronic pain.

The new targeted network involves two main structures of the basal ganglia, the subthalamic nucleus (STN) and the substantia nigra pars reticulata (SNr) in relation to a primary nociceptive structure from the brainstem, the lateral parabrachial nucleus (LBN). This network has been mainly studied in the context of pain symptoms in Parkinson's disease. The present work focused on the anatomical interconnectivity and involvement of this group of structure in the modulation of chronic pain.

This manuscript is dense and includes a huge amount of anatomical, behavioral and electrophysiological studies, with additional data in supplementary materials... The main striking discovery of this work is (i) the description of two new pathways, one between the STN and the PBN and the other between the SNr and the STN, and (ii) the demonstration of the role of both pathways to modulate the circuits electrophysiologically, and acute and chronic pain behaviorally.

The results can be subdivided into two main parts

Part 1: STN – LPB pathways

Experience 1: Role of STN in acute and chronic pain: STN neurons are hyperactive in acute and chronic pain with an increase of c-fos expression in acute and chronic pain models and present an exacerbated evoked firing rate in brain slices recordings.

Experience 2: In vivo investigation of STN dynamic changes in response to pain stimulation using AAV-CaMKII-GCaMP6f injection in the STN and GCaMP6f signal recordings. The results confirm results from experience 1 and suggest the involvement of STN Glutamateric neurons.

Experience 3: Dissection of the circuit underlying the involvement of the STN in painlike behavior with (i) Characterization of STN-PBN pathway using anterograde (Alexa488-conjugated WGA) and retrograde (not indicated in the paper, you have to look in the supplementary material) tracing; (ii) characterization of LPB neuronal types innervated by STN. 80 % are glutamatergic. The results confirm the existence of a new pathway between the notion yen STN and LPB.

Experience 4: Characterization of STN-LPB projection functionality using (i) AAVCaMKII-ChR2-eYFP injected into STN, optogenetic stimulation of STN infected neurons, and slice recording of LPB. LPB firing rate was increased by photostimulation of the STN and (ii) c-fos immunohistochemistry showing an increased number of c-fos positive neurons in the LPB. The results confirm STN ability to modulate LPB neuronal activity.

Experience 5: Exploration of STN-LPB pathway contribution to pain processing using optogenetic modulation (excitatory and inhibitory) targeting specifically this pathway using retro-virus. The aim is to test the effect of specific STN-LPB pathway on pain response. The results support that stimulation of STN-LPB neurons induces mechanical hypersensitivity. Inhibition of STN-LPB pathway had no effect.

Experiment 6: Test whether inactivation of STN-LPB pathway can ameliorate pain-like hypersensitivity. In acute and chronic pain states, using same technique as before but in acute and chronic pain models. The results show that silencing STN-LPB pathway regulates nociceptive responses behaviorally for acute pain but also in chronic pain states that involve central sensitization.

Experiment 7: Test whether the STN-LPB pathway modulate the emotional component of pain using pain-relief-conditioned place preference behavioral apparatus. The results show that silencing STN-LPB activity also attenuated the emotional components

of pain.

Experiment 8: To test whether the previous effects are due to a specific recruitment of STN-LPB pathway, without the contaminating involvement of axonal input to other nuclei. The author modulated STN terminals in the LPB using optogenetic and behavioral assessment. The author concluded that the activation of the STN-LPB projection is sufficient to reduce pain-threshold but play a limited role in maintenance of the normal pain threshold.

Experiment 9: Test the role of STN-PBN projection in modulation of acute and chronic pain.

From experiment 8 and 9: the authors concluded that STN-LPB projection control painlike hypersensitivity through axonal inputs to the LPB rather than through input to other regions outside the LPB. These experiments are not clear enough to understand the conclusion of the authors.

Experiment 10: To test if the STN-LPB glutamatergic projection is involved in pain-like behavior. The results suggest that disrupting neuronal activity in the STN is sufficient to reduce nociceptive responses in LPBGlu neurons.

Experiment 11: To test if the ability of STN to relieve pain may reflect cellular and/or synaptic changes within STN-LPB pathway since chronic pain has been associated with changes in the plasticity of the pain transmission pathways. They evaluated whether enhancement of the STN-LPB projection observed in neuropathic pain involves presynaptic or post-synaptic mechanisms using paired pulse ratio. The results show that the paired pulse ratio was reduced in LPB neurons in SNI mice compared to sham mice. The authors concluded of pre-synaptic mechanisms to enhance STN-LPB activity in a chronic pain model.

Part 2: The authors then looked for the origin of STN hyperactivity in acute and chronic

pain.

Experiment 12: Test whether GABAergic neurons activity in the Pedunculopontin nucleus (PPN), Globus pallidus external (GPE) and/or Substantia nigra pars reticulata (SNr) is modified in a neuropathic pain state. The authors found that SNr neurons, but not GPe or PPN neurons are hypoactive in SNI mice compared to sham mice, suggesting SNr attenuated inhibitory input to STN may represent a major contributor for STN hyperactivity.

Experiment 13: To characterize the anatomical and functional connectivity of the SNrGABA, STNglu and LPBGlu pathway. The results confirm that alteration of the SNr input to the STN modulates nociceptive responses in the LPB.

Experiment 14: to test whether SNrGABA neurons modulate the nociceptive responses in LPB via the STN.

Experiment 15: To test whether the ablation of STN prevents the pain-relieving effect of activation of the SNr-STN-LPB pathway.

Experiment 16: To test whether the stimulation of SNr-STN pathway has an impact on the emotional component of pain in a CPP paradigm. This summary was meant to illustrate the amount of data squeezed into a small format manuscript. There are two very important stories in this manuscript and the crucial discovery made by the authors are not properly highlighted in the present form. My suggestion would be to split the two parts of the results into two different articles.

This will allow to facilitate the reading of the manuscript and the results.

Part 1: A single manuscript on the STN-LPB pathway would help to clarify the aim and conclusion of the experiment 8 and 9 (see above). The authors should be careful to use the right terminology between nociception and pain. It would also be crucial to add a thorough histological analysis including for example all the injection sites of their viruses. The attempt to present all the data was done to some important methodological and histological disadvantage.

Part 2: This set of experiment needs clarification on several aspects: The authors have

focused their experiments of STN firing pattern control toward an inhibitory hypothesis. However, this control may be more complex than with GABA inputs only. For example, it has been shown that cortico – subthalamic pathway provide STN firing rate (Magill et al., 2000; Bolam et al., 2002) while this is a glutamergic input. Excitatory input to the STN may also modulate STN GABAergic interneuron to modulate STN Glu neuronal activity. Looking for the origin of STN hyperactivity may require more experiments to complete the story.

There is a misunderstanding for several anatomical pathways described by the authors. The authors cited two articles to justify GPe, SNr and PPN GABA input to the STN.

- Benarroch EE, Neurology, 2008

- Yang et al. Nature Neuroscience, 2021.

While GPe – STN pathway is indeed well known, the connection between the PPN and the STN are reciprocal and involves Glutamatergic and Cholinergic inputs (Benarroch, 2008), not GABAergic. The connection between SNr and STN is not described in the previous papers and to my knowledge has never been characterized. The SNr-STN pathway is not well known as claimed by the authors. This is why the results from part 2 are really interesting as it would represent another reciprocal loop similar to those described between the striatum and the GPe, and between the GPe and STN. This is also why I would suggest to the authors to thoroughly analyze this new circuit anatomically and functionally, physiologically and in the context of pain control in a separate manuscript.

Finally, STNGLU neurons are hyperactive and are well known to project to SNrGABA neurons. This hyperactivity should, in theory, lead to a hyperactivity of SNr neurons. How can the authors explain SNr hypoactivity with this connectivity?

Responses to comments from Reviewer 1

We appreciate the reviewer for carefully reading through the manuscript, thoroughly evaluating the rationality of each experiment, and raising professional critiques to help us improve the manuscript. Below, we listed the critiques the reviewer may want us to address.

1. Experiment 8: To test whether the previous effects are due to a specific recruitment of STN-LPB pathway, without the contaminating involvement of axonal input to other nuclei. The author modulated STN terminals in the LPB using optogenetic and behavioral assessment. The author concluded that the activation of the STN-LPB projection is sufficient to reduce pain-threshold but play a limited role in maintenance of the normal pain threshold.

Experiment 9: Test the role of STN-PBN projection in modulation of acute and chronic pain.

From experiment 8 and 9: the authors concluded that STN-LPB projection control painlike hypersensitivity through axonal inputs to the LPB rather than through input to other regions outside the LPB. These experiments are not clear enough to understand the conclusion of the authors.

Response: The reviewer raised an important issue in these two experiments. To understand whether LPB-projecting STN neurons innervate other nuclei, we injected AAV-CaMKII-Cre-eGFP in the STN and AAV retro-hSyn-DIO-ChR2-mCherry in the LPB neurons. This strategy allows labeling of LPB-projecting STN (STN-LPB) neurons (Supplementary Fig. 3A, 3B). We then imaged brain sections and found the densest eYFP-labeled fibers in the LPB (Supplementary Fig. 3I-3K). Besides the LPB, STN-LPB neurons project to the substantia nigra pars compacta and reticulata (SNc and SNr), pedunculopontine nucleus (PPN), ventral tegmental area (VTA), globus

pallidus interna and externa (GPi and GPe), and ventral pallidum (VP). This evidence support that the reviewer's concern is rational. Indeed, we observed that stimulating STN-LPB neurons in the STN and the STN-LPB projection in the LPB resulted in different modulation of mechanical and / or thermal threshold on unilateral or both hind paws (Fig. 3B, 3C, 5B). These results hint that manipulating the STN-LPB projection in the LPB is a better way to interrogate the role of this projection in pain modulation. Therefore, we draw conclusion using the data acquired with this strategy.

For these data, we wrote in Discussion:

“We observed an inconsistent phenomenon that stimulation of STN-LPB neurons regulated mechanical but not thermal threshold on both sides, but stimulation of the STN-LPB projection regulated both mechanical and thermal thresholds on the contralateral side. We postulate that stimulation of STN-LPB neurons may affect other pathways besides the STN-LPB projection (Supplementary Fig. 3I – 3K), probably leading to different effects on pain modalities on either side.”

2. The authors should be careful to use the right terminology between nociception and pain.

Response: We thank the reviewer for this critique. We corrected the misused terminology in the manuscript.

3. It would also be crucial to add a thorough histological analysis including for example all the injection sites of their viruses.

Response: We improved presentation of our histological data to confirm the viral expression and the positions of optical fiber implants. In Methods, we wrote:

“Viral expression (Fig. 2F, 2G, 6B, 6F, 6M, 7F; Supplementary Fig. 3C, 3G, 5B, 7C, 7I, 10) and the position of fiber implants in each mouse were confirmed histologically after the experiments. We only included mice with viral expression confined to the SNr, STN, LPB, GPe or PPN and optical fiber placement in or above

the STN or LPB (Supplementary Fig. 1Q, 3R, 3S, 5L, 6X, 8I, 9U).”

4. The attempt to present all the data was done to some important methodological and histological disadvantage.

Response: We revised the main text, figure legends, and supplementary information to improve our description of methods, and added images and diagrams to overcome our histological disadvantage in data presentation.

5. There are two very important stories in this manuscript and the crucial discovery made by the authors are not properly highlighted in the present form. My suggestion would be to split the two parts of the results into two different articles.

Response: We appreciate the reviewer’s suggestion. In another study (Yin et al., 2022), we examined the role of SNr neurons in the modulation of pain-like behaviors in both physiological and pain states using optogenetics, neuronal tracing, chemogenetics, and neuropharmacological methods, as well. In that study, we demonstrated the existence of the functional SNr-STN projection. Pharmacologically blocking this projection eliminated analgesic effects of SNr neuron activation. The manuscript was published online in the journal *Pain* on Jan 25, 2022 (10.1097/j.pain.0000000000002588), after we submitted the present study to *Nature Communications*. Thus, we added new experiments other than we included in the accepted manuscript to address the SNr-STN projection.

In revised manuscript, we remained the data about the function of SNr-STN-LPB pathway because we wondered whether the STN-LPB projection is regulated by the SNr-STN projection and whether the effect is sufficient to modulate pain. The data in the present study add a downstream circuit for the SNr-STN projection. In revision, we changed our wording to facilitate the flow of the story.

6. There is a misunderstanding for several anatomical pathways described by the authors. The authors cited two articles to justify GPe, SNr and PPN GABA input to the

STN.

- Benarroch EE, *Neurology*, 2008

- Yang et al. *Nature Neuroscience*, 2021.

While GPe – STN pathway is indeed well known, the connection between the PPN and the STN are reciprocal and involves Glutamatergic and Cholinergic inputs (Benarroch, 2008), not GABAergic.

Response: We thank the reviewer for correcting our misunderstanding of the literature. We revised this part. In the present study, we mentioned the PPN as a potential GABAergic input nucleus of the STN neurons. When we labeled PPN GABAergic neurons with optogenetics, we performed patch-clamp techniques and detected photo-IPSCs in STN neurons. The data support that STN neurons may receive GABAergic inputs from PPN GABAergic neurons. It is possible that STN neurons may receive GABAergic inputs from the PPN. But whether the inputs are weaker than glutamatergic and cholinergic inputs from the PPN warrants further confirmation.

About this statement, we wrote in Result:

“Based on our recent study demonstrating the SNr-STN projection confers an analgesic effect in several pain conditions ¹, we hypothesized that compromised inhibitory synaptic inputs to STN neurons may be associated with their hyperactivity in various pain states. Indeed, we found a reduced frequency but not amplitude of spontaneous inhibitory postsynaptic currents (sIPSCs) in STN neurons, compared with those in sham mice (Fig. 6H, 6I and 6J). We then combined optogenetics and brain slice patch-clamp recordings to address whether major and potential GABAergic input nuclei of the STN are modified in the neuropathic pain state, contributing to the hyperactivity of STN neurons. First, we injected AAV-DIO-ChR2-eYFP into the GPe, SNr, or PPN of Vgat-Cre mice to label local GABAergic neurons (Fig. 6K, 6L, Supplementary Fig.7A, 7B, 7G and 7H). Optogenetic activation of GABAergic terminals from the SNr, GPe, or PPN in the STN elicited IPSCs (photo-IPSCs) in STN neurons in brain slices and these photo-IPSCs were blocked by bicuculline (BIC, 10 μ M), a GABA_A receptor antagonist (Fig. 6M, Supplementary Fig. 7C and 7I). Mice

expressing Chr2-eYFP in GABAergic neurons in the SNr (SNr^{GABA}), GPe (GPe^{GABA}), or PPN (PPN^{GABA}) were subjected to contralateral SNI or sham surgery. We found that the spontaneous and evoked firing rates were significantly reduced in SNr^{GABA} neurons, but not in GPe^{GABA} or PPN^{GABA} neurons in week 2 after SNI surgery relative to sham mice (Fig. 6N – 6P, Supplementary Fig. 7D – 7F, 7J – 7L). There was no difference in resting membrane potential and membrane input resistance in SNr^{GABA} neurons, GPe^{GABA} or PPN^{GABA} neurons between sham and SNI mice (Supplementary Table 2). These data suggest that hypoactivity of SNr^{GABA} neurons in SNI mice may contribute to the attenuated inhibitory input to STN neurons.”

7. For example, it has been shown that cortico – subthalamic pathway provide STN firing rate (Magill et al., 2000; Bolam et al., 2002) while this is a glutamergic input. Excitatory input to the STN may also modulate STN GABAergic interneuron to modulate STNGlu neuronal activity. Looking for the origin of STN hyperactivity may require more experiments to complete the story.

Response: We admit that we need to thoroughly examine pain-related changes in both excitatory and inhibitory inputs to the STN to find the synaptic mechanisms underlying the hyperactivity of STN neurons in pain states. Considering that the strength of GABAergic inputs is critical to reset the firing rate and firing pattern in STN neurons² and that stimulation of SNr GABAergic neurons sufficiently inhibits STN neurons and mitigates pain-like hypersensitivity in pain states¹, in the present study, we focused on GABAergic inputs. Considering that this manuscript aims to address the role of SNr-STN-LPB projection in pain modulation, adding glutamatergic inputs to the STN here may cause distraction of the main point. We will address the modification of the glutamatergic inputs to the STN in our future study.

In addition to glutamatergic projection neurons, the STN contains a small population of GABAergic interneurons³. This neuron type was well defined in morphology, but its function has largely been unknown. Future investigations are warranted to address whether these interneurons are involved in pain modulation, motor control, or other

physiological or pathophysiological functions.

8. *Finally, STNGLU neurons are hyperactive and are well known to project to SNrGABA neurons. This hyperactivity should, in theory, lead to a hyperactivity of SNr neurons. How can the authors explain SNr hypoactivity with this connectivity?*

Response: We discussed this very important issue in Discussion. We wrote:

“Previous studies have reported that SNr neurons respond diversely to various pain modalities and that using pharmacological or electrical neuromodulation to either inhibit or stimulate the SNr produces analgesic effects ⁴. A recent study demonstrated that SNr neurons receiving direct nociceptive signals from excitatory neurons in the LPB are essential for pain perception ⁵. Similarly, we previously reported that SNr neurons innervated by the STN partially mediate hyperalgesia in parkinsonian mice ⁶. Contrary to this, but similar to another study ¹, we observed that stimulation of SNr neurons led to analgesic effects. These results suggest that SNr neurons are diverse in terms of their involvement in pain perception, depending on which neural pathway they belong to. That is, those innervated by the LPB and STN may facilitate the transmission of pain signals, whereas those innervating the STN suppress pain perception. Indeed, two subsets of SNr neurons that respectively innervate STN neurons and are innervated by STN neurons are largely separated ¹. Taking the previous studies together with our current finding, we propose that SNr^{GABA} neurons receive pain-related information from the LPB and, in turn, regulate the downstream STN-LPB pathway and control pain responses. However, in pain states, SNr^{GABA} neurons are hypoactive and lose the ability to initiate feedback inhibition of the STN–LPB pathway, resulting in hyperalgesia.”

Reviewer 2

In this manuscript titled “A nigro–subthalamo–parabrachial pathway modulates 1 pain-like behaviors”, the authors explored the involvement of substantia nigra pars reticulata (SNr)- subthalamic nucleus (STN)- lateral parabrachial nucleus (LPB) pathway in acute mice inflammation and neuropathic pain models. The authors proved that STNGlu neurons are activated after injury and can be activated by peripheral von Frey fiber stimulation (Figure 1), STNGlu neurons directly innervate LPBGlu neurons (Figure 2), optogenically activation of STNGlu neurons can induce hypersensitivity to non-noxious von Frey stimulation in naïve rats while optogenically inactivation of STNGlu neurons takes analgesic effects in mice with painful injury (Figure 3), optogenically inactivation of terminals of STNGlu neurons in LPB also takes the similar analgesic effects (Figure 4), chemogenetically inhibition of STN reduced the response of LPBGlu neurons to mechanical stimulation by von Frey fibers in naïve rats (Figure 5). Then they found that there are increased activity of STNGlu and LPBGlu neurons, and decreased activity of SNrGABA neurons (Figure 6). They further prove the SNrGABA–STNGlu–LPBGlu pathway (Figure 7). Finally, they selectively ablated STN neurons, which prevented the pain-relieving effect of activation of the SNr–STN–LPB pathway (Figure 8). They concluded that this SNrGABA- STNGlu- LPBGlu is critical for the development of mechanical and thermal hypersensitivity in acute and ‘chronic’ pain states. This is an interesting and well-designed study, which provides valuable findings to this field. I have the followings suggestions/ questions.

1. The authors proved the SNrGABA–STNGlu–LPBGlu pathway (Figure 1-7). Then, they selectively ablated STN neurons, which prevented the pain-relieving effect of activation of the SNr–STN–LPB pathway (Figure 8). If STNGlu neuronal activity is critical in the development of mechanical and thermal hypersensitivity after painful nerve injury, is there any explanation on that those mice with ablation of STNGlu neurons still developed hypersensitivity to von Frey and heat after nerve injury (Figure

8), which indicates that activity of STN^{Glu} neurons is not critical in the pain development and is contradictory to their main point? It seems that this pathway is only one of many regulatory pathways in the brain in pain condition.

Response: We agree with the reviewer that the STN is not an essential component for pain development, but it is an effective pain modulator. In Discussion, we wrote:

“Interestingly, the mice with selective ablation of STN-LPB neurons still developed mechanical and/or thermal hypersensitivity after capsaicin injection or SNI surgery (Fig. 8). These data demonstrate a modulatory role of STN-LPB pathway in acute and persistent pain conditions.”

2. Regarding the concept of acute pain and chronic pain, 7 days after SNI should be still in acute phase of neuropathic pain, even though it is debatable on the timing of transition from acute to chronic pain. According to a study (PMID: 34244365), there is dynamic change of neuronal activity after nerve injury. Please correct this misconception about chronic pain in the whole manuscript.

1 day after Complete Freund's adjuvant (CFA) paw injection induced pain cannot also be classified as chronic pain, as they claimed in Figure S1-B.

Response: We thank the reviewer for pointing out the mistakenly used terminology. For SNI mice, we changed “chronic pain” to “persistent pain”. For CFA mice, we changed “chronic pain” into “persistent inflammatory pain”.

3. Regarding the concept of CPP test, CPP test is not an indicator of emotional component of neuropathic pain. Any effective analgesic treatment can induce CPP, even it only targets peripherally. So, CPP positive is only an indicator of reduced spontaneous pain, not emotional component of pain. Please correct it thoroughly.

Response: We thank the reviewer for correcting our misinterpretation of the data about

CPP test.

In Results, we wrote: “Pain-relief effect of optogenetic inhibition of STN-LPB neurons was further examined in the conditioned place preference (CPP) test (Fig. 3L), an operant behavioral test based on the fact that pain relieving treatments are rewarding and produce a CPP in mice with nerve injury^{7,8,9}.”

In Discussion, we wrote “Spontaneous ongoing pain is one major symptom presented in patients with neuropathic pain. The CPP paradigm has been used to evaluate whether a treatment mitigates spontaneous pain⁸. We observed that stimulation of the SNr–STN projection and inhibition of the STN–LPB projection established a CPP in SNI-NpHR mice but inhibition of STN-LPB neurons did not induce CPP in Sham-NpHR mice, suggesting that these manipulations alleviate spontaneous pain, but do not cause rewarding effect (Fig. 3). These data indicate that, in addition to modulating pain thresholds, the SNr^{GABA}–STN^{Glu}–LPB^{Glu} pathway regulates spontaneous pain in neuropathic pain condition.”

4. It is not clear what the purpose of experiments of GCaMP6f without injury (Fig. 1I-P) is, even though it was claimed to test “the dynamic change of the activity of STN neurons in response to pain stimulation”. Which part of the data represent the dynamic change of the activity? Mechanical stimulation with von Frey fibers is non-noxious. It is irrelevant to pain or nociception. They concluded that “these results suggest that STNGlu neurons encode nociception”, which is not true as activated neurons after nerve injury doesn’t mean they encode nociception. Their ablation data (Figure 8) also confirmed this that STNGlu neurons are not critical in encode painful sensation.

Response: We agree with the reviewer. To answer the question whether STN neurons respond to noxious and non-noxious stimulus, we did additionally experiments. First, we observed that GCaMP6 signal in STN neurons were enhanced in response to 48 °C heating, a noxious stimulus, of hind paws (Fig. 1). Second, we found that GCaMP6 signals in STN neurons were enhanced upon suprathreshold stimulus, while the

responses became more robust in capsaicin- and SNI-induced pain-like behaviors (Supplementary Fig. 1). Therefore, STN neurons respond to both non-noxious and nociceptive stimulus. Our optogenetic and chemogenetic modulation experiments reveal that STN-LPB projection modulate pain-like behaviors.

5. Details on electrophysiological recordings are needed. Are those recordings on EPSC and IPSC recorded with GABAAR blockers or NMDAR and AMPAR blockers, respectively? What is the holding potential? What is reason that spontaneous IPSC (Fig. 6G-I), not miniature IPSC, was recorded to explore the pre- vs post- synaptic mechanisms?

On the recording of firing in STN (Figure S1D-G) and LPB neurons (Fig. 6C, D), what is duration of the current injection? What is the normal resting membrane potential of these neurons? What is the standard to exclude recordings as it seems that many neurons recorded have abnormal resting membrane potentials, which is an indicator of bad recordings with too much leaking and contribute to the results dramatically especially in current clamp whole-cell recording?

Response: We added these information in the manuscript and in the supplementary information.

In supplementary information, we wrote:

“In our recording condition, the reversal potential of Cl⁻ is -81 mV. When the neurons were held at -50 mV, excitatory and inhibitory postsynaptic currents (EPSCs and IPSCs) were respectively inward and outward currents, respectively. Therefore, EPSCs and IPSCs are identifiable without pharmacological isolation (see our our previous studies ^{1, 6, 10}). In the present study, AP-5 (50 μM) plus CNQX (20 μM) and bicuculline (10 μM) was applied through bath perfusion to verify glutamatergic or GABAergic properties of the postsynaptic currents, respectively. In the revised manuscript, we included the information of the resting membrane potential and input resistance of all recorded neurons from sham and SNI mice in Supplementary table 1 and 2. Firing rate evoked by current injections (1s, 20 – 200 pA) were recorded in the

current-clamp mode. After formation of whole-cell recordings, the neurons with a holding current bigger than -50 pA and with a resting membrane potential more depolarized than -40 mV were not included in the analysis.”

There are at least three reasons we recorded sIPSCs but not mIPSCs from STN neurons. First, in our previous recordings¹¹, we observed that in the STN, TTX does not affect both amplitude and frequency of sIPSCs and sEPSCs. Second, significant changes of amplitude and frequency of both sIPSCs and mIPSCs hint the involvement of postsynaptic and presynaptic mechanisms, respectively. Third, sIPSCs, but no mIPSCs, are action-potential dependent, and enhancement or inhibition of the activity of the presynaptic neurons leads to an increase or a reduction of sIPSC frequency. In the present study, we aim to explore whether GABAergic synaptic inputs are modified in neuropathic pain. Therefore, sIPSCs may be suitable to answer this question.

6. SNr, STN, and LPB innervate each other in both directions. SNr receive innervations from STN and LPB, and send innervations to them. As the authors mentioned in the discussion “we propose that SNr GABA neurons receive pain related information from the LPB”, shouldn't increased LPB neuronal activity after painful nerve injury will activate SNrGABA neurons, thus reduce STN activity and then reduce LPB activity? SNr neurons also receive innervations from STN. If STN neurons and LPB neurons are all activated after painful nerve injury, SNr neurons should be also activated, not deactivated, to provide inhibitory feedback regulation.

Response: We admit that the reviewer's hypothesis very likely holds true. From our previous studies, we realize that SNr neurons are heterogeneous in terms of synaptic inputs and outputs. We tried to address this important issue in Discussion.

We wrote:

“Previous studies have reported that SNr neurons respond diversely to various pain modalities and that using pharmacological or electrical neuromodulation to inhibit or stimulate the SNr produces analgesic effects⁴. A recent study demonstrated that SNr neurons receiving direct nociceptive signals from excitatory neurons in the LPB are

essential for pain perception ⁵. Similarly, we reported that SNr neurons innervated by the STN partially mediate hyperalgesia in parkinsonian mice ⁶. Contrary to these studies ^{5,6}, but similar to another study ¹, we observed that stimulation of SNr neurons led to analgesic effects. These results suggest that SNr neurons are diverse in terms of their involvement in pain perception, depending on which neural pathway they belong to. That is, those innervated by the LPB and STN may facilitate the transmission of pain signals, whereas those innervating the STN suppress pain perception. Indeed, two subsets of SNr neurons that respectively innervate STN neurons and are innervated by STN neurons are largely separated ¹. Taking the previous studies together with our current findings, we propose that SNr^{GABA} neurons receive pain-related information from the LPB and, in turn, regulate the downstream STN-LPB pathway and control pain responses. However, in pain states, SNr^{GABA} neurons are hypoactive and lose the ability to initiate feedback inhibition of the STN–LPB pathway, resulting in hyperalgesia.”

7. Please provide reference for this sentence “Contrary to this, we observed that stimulation of SNr neurons led to analgesic effects” in page 16, line 406.

Response: We thank the reviewer for pointing out this. We added the references. The revised statement is

“A recent study demonstrated that SNr neurons receiving direct nociceptive signals from excitatory neurons in the LPB are essential for pain perception ⁵. Similarly, we previously reported that SNr neurons innervated by the STN partially mediate hyperalgesia in parkinsonian mice ⁶. Contrary to these studies^{5,6}, but similar to another study ¹, we observed that stimulation of SNr neurons led to analgesic effects. These results suggest that SNr neurons are diverse in terms of their involvement in pain perception, depending on which neural pathway they belong to.”

Reviewer 3

The manuscript describes a hitherto uncharacterized pathway that may regulate pain perception, from the subthalamic nucleus (STN) to the parabrachial nucleus (PB). This complements the previous characterized reciprocal pathway from PB to STN (which the authors, surprisingly, do not explicitly discuss). The authors apply an impressive array of behavioral, electrophysiological and functional imaging approaches to test the hypothesis that the STN->PB pathway is causally related to both acute and chronic pain. The findings are novel and of significance. However, a number of concerns reduce the rigor of the findings and their conclusion.

1. That only male mice were used in a study of pain is inexcusable. For at least two decades the pain field has emphasized the stark sex differences in pain processing, and the clinically significant differences in signs and symptoms and in therapeutic approaches to pain. Studying only males should no longer be accepted.

Response: To mitigate this flaw, we examined the role of the STN-LPB projection and the SNr-STN-LPB projection in the modulation of pain-like behaviors in female mice (Supplementary Figure 6 and Figure 9). We observed that these projections exerted similar effects on pain-like behaviors in male and female mice.

In Results, we wrote the following two paragraphs.

“We also found that the STN-LPB pathway regulates pain-like behaviors in female mice under different pain conditions (Supplementary Fig. 6A) similarly to its effects on in male mice (Fig. 5 and Supplementary Fig. 5). We observed that unilateral optogenetic stimulation of the STN-LPB pathway decreased both mechanical and thermal pain thresholds on the contralateral but not the ipsilateral hind paw in naïve female mice (Supplementary Fig. 6B, 6C); on the contrary, optogenetic inhibition of this pathway had no effect on mechanical and thermal threshold on both hind paws (Supplementary Fig. 6D, 6E). In capsaicin-, CFA-, and SNI female mice, optogenetic inhibition of the

STN-LPB pathway mitigated pain-like behaviors on the contralateral but not the ipsilateral hind paw (Supplementary Fig. 6F – 6T). CPP was developed in SNI mice by optogenetic inhibition of this pathway (Supplementary Fig. 6U – 6W). These results indicate that the effects of STN-LPB pathway on pain-like behaviors may not have sex-differences.”

“Moreover, we examined the modulatory effect of the SNr–STN–LPB pathway on physiological pain thresholds and pain-like hypersensitivity in pathological pain states in female mice (Supplementary Fig. 9A–9C). Consistent with those in male mice (Fig. 8; Supplementary Fig. 8), optogenetic stimulation of the SNr–STN–LPB pathway did not change mechanical and thermal pain threshold, motor function in the open field test, and performance in the adhesive tape removal test in naïve female mice (Supplementary Fig. 9D–9G, 9P–9T), but mitigated mechanical hypersensitivity in capsaicin female mice (Supplementary Fig. 9H – 9J). Female mice with SNI also showed attenuated mechanical and thermal hypersensitivity and developed CPP upon optogenetic stimulation of this pathway (Supplementary Fig. 9K – 9O). These data suggest that the STN^{Glu}–LPB^{Glu} pathway also regulates pain-like hypersensitivity in female mice.”

2. This (line 319) is a head scratcher: Ablating STN neurons "greatly decreased the number of neurons in the STN (Fig. 8C and 8D); however, baseline nociceptive sensitivity and locomotion were not affected (Fig. S7B – S7H)." Does this finding not disprove the overarching hypothesis of this study?

Response: This finding is consistent with our previous studies (Luan et al., 2020; Yin et al., 2022). In these studies, we inhibited STN neurons with either NpHR-mediated optogenetic cellular inhibition or optogenetic stimulation of the SNr-STN GABAergic projections and observed no modulation of mechanical and thermal threshold in naïve mice. On the contrary, these manipulations increased mechanical and thermal threshold in pain states. Our explanation is that the STN is an important modulatory nucleus in pain processing. It may not contribute to the maintenance of normal pain threshold, but it may interact with pain-related nuclei to confer central sensitization in pain states, in which inhibition of STN neurons effectively mitigates the central sensitization.

3. *In all experiments comparing data from sham and treated (e.g. SNI, CFA) animals, data (e.g. electrophysiological metrics) should be averaged from each animal, and animals (as opposed to neurons) should be used as the counting objects for statistical analyses. The same holds also for analyzing cFos data, where animals, and not sections, should be counted. This applies, for example, to Figs. 1b,d,f; S1e,f,g; S3; S4; S6' 6, and many more.*

Response: We averaged each parameter obtained from each mouse, used the average to quantify the parameter in each mouse, and changed all sample sizes into numbers of mice. See data points in the Figure and supplementary figures, and descriptions in Figure legends of main figures and supplementary figures.

4. *There is no mention of an a-priori power analysis, or any explicit consideration of statistical power.*

Response: In Methods and Materials section, we added “The number of mice used in each experiment was calculated in a priori power analysis (StatMate 2.0), the power of each experiment was set to 0.8.”.

5. *The authors appear to acknowledge that manipulating STN is likely to affect motor behaviors, including reflexive measures of nociception and non-reflexive pain behaviors. Relying only on locomotion, assessed in open field, is likely not sufficient to allay this concern.*

Response: We agree with the reviewer that the motor function involved in paw withdrawal may differ from that in locomotion. In the present study, we examined the effect of optogenetic manipulation of the STN-LPB projection and the SNr-STN-LPB projection on tape removal in mice (Supplementary Figure 5 and 9). As tape removal involves both aversive sensation and motor reflex, it may provide information about whether sensory and motor function, important for pain reflex, is changed by the manipulation. We observed that manipulation of the STN-LPB projection and the SNr-STN-LPB projection did not change the ability to remove the sticky tape attached on

the hind paw.

6. Preclinical models of neuropathic pain, such as SNI used here, do not consider one week post-surgery as the "chronic pain" period. The 7 to 10 days post-SNI period used to assess behavior and electrophysiology in some experiments needs to be justified. The authors should also justify why in some experiments they focused on 7 days post-SNI, and in others they used a 2 weeks post-SNI period.

Response:

We assessed the role of the SNr-STN-LPB pathway in SNI induced neuropathic pain at 2 time points, mid-term (>7 days) and long-term (>4 weeks)^{12, 13, 14}. Because sensory hypersensitivities appear within 3 days after nerve injury. During mid-term, pain thresholds remain constantly low, therefore, pain thresholds, motor function, c-Fos staining, and patch-clamp recordings in SNI or sham mice were tested > 7 days, mostly between 2 and 3 weeks after SNI. We found that hyperactivity of STN neurons persisted up to 5 weeks after SNI (Supplementary Fig. 1), suggesting pain like behaviors were concomitant with alterations of neuronal activity in the STN. In most literature, CPP was used for measuring ongoing pain or emotional pain in neuropathic pain^{7, 8, 13, 15, 16, 17}. As emotional component of pain in neuropathic pain rodent models arise several weeks after surgery^{14, 18}, consistent with previous studies^{15, 16, 17}, CPP in present study was performed during long-term (4 weeks after surgery).

7. The hypothesis that STN activity is increased in persistent pain states is tested only using cFos labeling (Fig. 1). Why did the authors apply their GCaMP imaging approach to study only acute pain, and not also persistent pain states?

8. Related to that, micrographs showing GCaMP expression in STN would be informative.

Response: This is a reasonable concern. Both c-Fos labeling and GCaMP6 imaging have been widely used in neuroscience community to evaluate the changes of neuronal activity. But either has limitations. The c-Fos-labeling is not sensitive to transient enhancement of activity, but is suitable to detect neuronal activation lasting for a

considerable period of time, hours or days. On the other hand, GCaMP6 imaging has advantages in revealing a transient increase or decrease in neuronal firing, but it may not accurately reflect a gradual increase of neuronal activity in hours or days. In literature, GCaMP6 imaging has been rarely used to detect changes of neuronal activity which develop in hours or days. Therefore, we did not compare GCaMP6 signals before and during persistent pain states.

In revision, we examined the responses of STN neurons to suprathreshold mechanical stimulation in capsaicin-injected and SNI mice, and observed that STN-LPB neurons in these pain states responded more strongly than those in naïve mice.

9. Page 6, line 140: Provide a more descriptive detail of the finding, beyond stating “affect the activity of LPB neurons”.

Response: We changed the sentence into “These experiments demonstrate that optogenetic stimulation of STN glutamatergic terminals is sufficient to increase the activity of LPB neurons.”

10. Quantification in Fig. 2 is lacking or unclear. How many cells do the percentages represent? How many animals were assayed?

Response: We provided the information in Fig. 2. We counted 1070 WGA-labeled neurons in the LPB sections from 4 mice and 850 neurons were co-labeled with eYFP.

11. The finding that mechanical, but not thermal responses were affected by modulating STN activity appears contrary to previous findings, including from the authors' lab. This should be discussed.

Response: Thanks for reviewer’s concern. In our previous study⁶, we found that basal mechanical and thermal thresholds were decreased with optogenetic activation of STN neurons through different projections in naïve mice. These results suggest that STN neurons are diverse in terms of their involvement in different pain modalities, depending on which neural pathway they belong to. Indeed, we found in the present study that selective activation of LPB-projecting STN neurons decreased mechanical

but not thermal threshold. Furthermore, we observed that STN-LPB neurons also project to the substantia nigra pars compacta and reticulata (SNc and SNr), pedunclopontine nucleus (PPN), ventral tegmental area (VTA), globus pallidus interna and externa (GPi and GPe), and ventral pallidum (VP) (Supplementary Fig. 3I – 3K). We postulate that stimulation of LPB-projecting STN neurons may affect other pathways besides the STN-LPB projection, probably leading to different effects on pain modalities on either side.

12. Fig. 3(l-m): Please clarify what "recovery" refers to. For example, how long after optical activation?

Response: “recovery” in Fig. 3 refers to at least 2 hours after the discontinuation of optogenetic activation.

13. Figure 3(o-r) It appears that only SNI animals were used. The absences of sham animals should be justified. Equally importantly, the data in P and R panels should be compared directly using a 2-way ANOVA, and not treated as different statistical tests. This applies also to the data in Figure S4.

Response: We admit that sham animals are the corresponding control for SNI mice. We conducted the experiment in sham mice and included the data in Fig. 3R-T.

In Result, we wrote

“SNI mice with NpHR expression in STN-LPB neurons showed a preference to the yellow light conditioned chamber (Fig. 3O – 3T). Such preference was not seen in sham mice or SNI mice with mCherry expression in STN-LPB neurons. Our observation that light stimulation did not reduce the traveling velocity in the light stimulation-paired chamber (Fig. 3Q, 3T) excludes the possibility that altered locomotion interfered with the latency to shuttle between chambers.”

For statistical analysis, we used two-way ANOVA to analyze those data and revised figure legends accordingly.

14. The absence of sham controls for the data in Fig. 4 should be justified.

Response: We switched the order of Fig. 4 and Fig. 5 in last version. Thus, Fig. 4 is Fig. 5 in this version. We have several reasons for not having sham mice as controls for SNI mice. First, the purpose of the experiment is to test whether inhibition of the STN-LPB pathway mitigates pain-like hypersensitivity in SNI mice. We used eYFP-SNI mice as controls for NpHR-SNI mice to exclude the non-specific effect of yellow light illumination. Second, we observed that sham mice have similar mechanical and thermal thresholds as naïve mice. In naïve mice, we found no effect of optogenetic inhibition of the STN-LPB projection on mechanical and thermal thresholds (Fig. 5C). Therefore, we did not use sham control in this part of experiment.

15. The absence of CNO-only controls (without hM4Di) for the data in Figs. 5,7 should be justified.

Response: This is a good suggestion to eliminate the confounding effects of CNO on STN or SNr neurons by modulating targets other than hM4Di or hM3Dq. We performed patch-clamp recordings and observed significant effects of CNO on the firing rate of STN neurons expressing hM4Di and SNr neurons expressing hM3Dq, but not those without hM4Di and hM3Dq (Supplementary Fig. 10A-10F). Therefore, we did not add a group of mice subjected to injection of AAV-EF1 α -DIO-GCaMP6f in the LPB and injection of AAV-CaMKII-mCherry in the STN.

In Figure 5, we explored whether inhibition of STN neurons can modulate LPB GCaMP6 signals in response to mechanical stimulation, therefore, ip injection of saline was used as control of CNO injection in mice with hM4Di expression in STN neurons, and eYFP expression was used as control of GCaMP6 in the LPB. In this way, we recorded LPB-GCaMP6 signals after CNO or saline injection, and LPB-eYFP signals after CNO or saline injection. Similarly, in Figure 7, we explored the effect of SNr-STN neurons on LPB GCaMP6 signals. We used saline injection as control of CNO injection.

16. A graphical representation of experimental timelines for each experiment would be useful.

Response: We add experimental timelines in Fig. 2A, 2D, 3A, 3L, 5A, 6A, 6K, 8B, 8J, and supplementary Fig. 6A, 6R, 9B.

1. Yin C, Jia T, Luan Y, Zhang X, Xiao C, Zhou C. A nigra-subthalamic circuit is involved in acute and chronic pain states. *Pain*, (2022).
2. Wilson CJ, Bevan MD. Intrinsic dynamics and synaptic inputs control the activity patterns of subthalamic nucleus neurons in health and in Parkinson's disease. *Neuroscience* **198**, 54-68 (2011).
3. Emmi A, Antonini A, Macchi V, Porzionato A, De Caro R. Anatomy and Connectivity of the Subthalamic Nucleus in Humans and Non-human Primates. *Frontiers in neuroanatomy* **14**, 13 (2020).
4. Chudler EH, Dong WK. The role of the basal ganglia in nociception and pain. *Pain* **60**, 3-38 (1995).
5. Yang H, *et al.* Pain modulates dopamine neurons via a spinal-parabrachial-mesencephalic circuit. *Nature neuroscience*, (2021).
6. Luan Y, *et al.* Reversal of hyperactive subthalamic circuits differentially mitigates pain hypersensitivity phenotypes in parkinsonian mice. *Proc Natl Acad Sci U S A* **117**, 10045-10054 (2020).
7. Navratilova E, Xie JY, King T, Porreca F. Evaluation of reward from pain relief. *Ann N Y Acad Sci* **1282**, 1-11 (2013).
8. King T, *et al.* Unmasking the tonic-aversive state in neuropathic pain. *Nature neuroscience* **12**, 1364-1366 (2009).
9. Dalm BD, Reddy CG, Howard MA, Kang S, Brennan TJ. Conditioned place preference and spontaneous dorsal horn neuron activity in chronic constriction injury model in rats. *Pain* **156**, 2562-2571 (2015).
10. Zhou C, *et al.* Bidirectional dopamine modulation of excitatory and inhibitory synaptic inputs to subthalamic neuron subsets containing alpha4beta2 or alpha7 nAChRs. *Neuropharmacology* **148**, 220-228 (2019).
11. Xiao C, *et al.* Nicotinic receptor subtype-selective circuit patterns in the subthalamic nucleus. *The Journal of neuroscience : the official journal of the Society for Neuroscience*

35, 3734-3746 (2015).

12. Llorca-Torralba M, *et al.* Pain and depression comorbidity causes asymmetric plasticity in the locus coeruleus neurons. *Brain : a journal of neurology* **145**, 154-167 (2022).
13. Zhang Z, Gadotti VM, Chen L, Souza IA, Stemkowski PL, Zamponi GW. Role of Prelimbic GABAergic Circuits in Sensory and Emotional Aspects of Neuropathic Pain. *Cell reports* **12**, 752-759 (2015).
14. Alba-Delgado C, Llorca-Torralba M, Mico JA, Berrocoso E. The onset of treatment with the antidepressant desipramine is critical for the emotional consequences of neuropathic pain. *Pain* **159**, 2606-2619 (2018).
15. Huang J, *et al.* A neuronal circuit for activating descending modulation of neuropathic pain. *Nature neuroscience* **22**, 1659-1668 (2019).
16. Sun L, *et al.* Parabrachial nucleus circuit governs neuropathic pain-like behavior. *Nat Commun* **11**, 5974 (2020).
17. Wang H, *et al.* Incentive-thalamic Circuit Controls Nocifensive Behavior via Cannabinoid Type 1 Receptors. *Neuron* **107**, 538-551 e537 (2020).
18. Bravo L, Llorca-Torralba M, Suarez-Pereira I, Berrocoso E. Pain in neuropsychiatry: Insights from animal models. *Neuroscience and biobehavioral reviews* **115**, 96-115 (2020).

REVIEWER COMMENTS

Reviewer #1 (Remarks to the Author):

I thank the authors for the work done on their manuscript.

I have two additional questions:

- WGA-Alexa555 is an anterograde and retrograde tracer. How do you differentiate the cell bodies labelled from both transport since the PBN does also project to the STN? How do you control that the labeled cell bodies are not only due to the retrograde labeling of PBN cells projecting to the STN ?

- We still have individual examples of the viruses injections only. A supplementary figure with the plotting of all injection sites is necessary.

Reviewer #2 (Remarks to the Author):

This a resubmission of the manuscript titled "A nigro-subthalamo-parabrachial pathway modulates 1 pain-like behaviors". The authors addressed most of my questions. I still have question on my previous question #4 that "They concluded that "these results suggest that STNGlu neurons encode nociception", which is not true as activated neurons after nerve injury doesn't mean they encode nociception. Their ablation data (Figure 8) also confirmed this that STNGlu neurons are not critical in encode painful sensation."

As those mice with ablation of STN neurons still show hypersensitivity to mechanical stimulation (Figure 8 F-I), it is clear that STNGlu neurons are not critical in encoding painful sensation. Their conclusion in line 78-80 that "Our findings implicate the malfunction of a tandem pathway (SNrGABA to STNGlu to LPBGlu) in the pathophysiology of pain states. Reducing the output in this circuit is a potential therapeutic strategy for pathological pain" is not true, as the output from STNGlu is blocked and there is still allodynia after injury. It seems that this pathway is not critical in pain development while manipulating activity of this pathway can modulate pain ascending or descending pathways to take somewhat analgesic effects, like a lot of other brain regions.

Reviewer #3 (Remarks to the Author):

I continue to be puzzled by how the diverse findings fit into a comprehensible hypothesis. Central to my confusion are findings that ablating (or inactivating) STN neurons has no effect on "sensitivity and locomotion", yet the authors show that STN neurons respond to nociceptive stimuli, are reciprocally connected with major pain centers, and their modulation affects responses to nociceptive inputs. Similarly, the differential effects on thermal and mechanical stimuli—contradicting the authors' own prior publications—have not been adequately addressed.

Related to these concerns, and in agreement with another reviewer, it might be advisable for the authors to parse the data into 2 or more manuscripts, with tighter rigor and rationale. In the process, moving much of the important data currently in the supplementary data section into the main manuscript might be beneficial.

The "sticky tape" experiment does not address the concern that manipulating STN affects reflexive pain metrics.

The absence of controls animals with CNO injections only, without DREADDs, continues to be a concern.

"Page 6, line 140: Provide a more descriptive detail of the finding, beyond stating "affect the activity of LPB neurons". This prior concern is not addressed, as no quantification is provided.

I do not understand the authors' response to reviewer 2 question regarding analysis of spontaneous rather than miniature synaptic events. "TTX does not affect both amplitude and frequency of sIPSCs and sEPSCs": What does this mean? "significant changes of amplitude and frequency of both sIPSCs and mIPSCs hint the involvement of postsynaptic and presynaptic mechanisms, respectively": This seems incorrect. "enhancement or inhibition of the activity of the presynaptic neurons leads to an increase or a reduction of sIPSC frequency"" this seems incorrect, too.

REVIEWER COMMENTS

Reviewer #1 (Remarks to the Author):

I thank the authors for the work done on their manuscript.

I have two additional questions:

1. WGA-Alexa555 is an anterograde and retrograde tracer. How do you differentiate the cell bodies labelled from both transport since the PBN does also project to the STN? How do you control that the labeled cell bodies are not only due to the retrograde labeling of PBN cells projecting to the STN?

Response: We thank the reviewer for pointing out this important issue. We agree with the reviewer that the potential pitfall of using WGA for neuronal tracing is that it is transported both anterogradely and retrogradely. There are a number of studies using WGA for either anterograde or retrograde tracing only or both (Crossland, 1985; Goshgarian and Buttry, 2014; Levy et al., 2017; Mantyh and Peschanski, 1983; Menet et al., 2003; Xu et al., 2021). However, the retrograde labeling by WGA, if any, may vitiate our conclusion. Therefore, we injected an anterograde trans-synaptic viral vector, scAAV1-hSyn-Cre, into the STN of Ai9 mice (a tdTomato reporter line). Our immunohistochemistry data show that about 75% of tdTomato-positive LPB neurons were labeled with a CaMKII antibody (Fig. 2D-G). The result supports that a majority of LPB neurons receiving STN projections are glutamatergic.

2. We still have individual examples of the virus injections only. A supplementary figure with the plotting of all injection sites is necessary.

Response: To address this important issue, we added panels in supplementary figures showing the plotting of the sites of virus injection and optical fiber implantations (see supplementary figures 1P-Q, 2H-I, 2L, 3R-S, 5C, 5M, 6D, 6T, 8K-L, 9I-K, 10D, and

10V).

Reviewer #2 (Remarks to the Author):

This a resubmission of the manuscript titled “A nigro–subthalamo–parabrachial pathway modulates 1 pain-like behaviors”. The authors addressed most of my questions. I still have question on my previous question #4 that “They concluded that “these results suggest that STN Glu neurons encode nociception”, which is not true as activated neurons after nerve injury doesn’t mean they encode nociception. Their ablation data (Figure 8) also confirmed this that STN Glu neurons are not critical in encode painful sensation.”

As those mice with ablation of STN neurons still show hypersensitivity to mechanical stimulation (Figure 8 F-I), it is clear that STN^{Glu} neurons are not critical in encoding painful sensation. Their conclusion in line 78-80 that “Our findings implicate the malfunction of a tandem pathway (SNr^{GABA} to STN^{Glu} to LPB^{Glu}) in the pathophysiology of pain states. Reducing the output in this circuit is a potential therapeutic strategy for pathological pain” is not true, as the output from STN^{Glu} is blocked and there is still allodynia after injury. It seems that this pathway is not critical in pain development while manipulating activity of this pathway can modulate pain ascending or descending pathways to take somewhat analgesic effects, like a lot of other brain regions.

Response: We appreciate the reviewer provides a rationale interpretation for our results. We agree with the reviewer that this pathway is important for the maintenance, instead of the development, of pain states. Therefore, we corrected our conclusion in Introduction “Our findings implicate the malfunction of a tandem pathway (SNr^{GABA} to STN^{Glu} to LPB^{Glu}) in the pathophysiology of pain states” to “We demonstrate that selective manipulation of the tandem pathway (SNr^{GABA} to STN^{Glu} to LPB^{Glu}) ameliorates pain-like behaviors in pain states.”

Reviewer #3 (Remarks to the Author):

1. I continue to be puzzled by how the diverse findings fit into a comprehensible hypothesis. Central to my confusion are findings that ablating (or inactivating) STN neurons has no effect on "sensitivity and locomotion", yet the authors show that STN neurons respond to nociceptive stimuli, are reciprocally connected with major pain centers, and their modulation affects responses to nociceptive inputs. Similarly, the differential effects on thermal and mechanical stimuli—contradicting the authors' own prior publications—have not been adequately addressed.

Response: We thank the reviewer for raising these two points.

It is known that a series of cellular and molecular processes are implicated in the initiation and maintenance of pain. Our evidence supports that STN neurons may be among the ones that modulate the maintenance of pain states. 1) STN neurons respond to nociceptive stimuli, suggesting their involvement in pain-like behaviors. 2) Inhibition of STN neurons attenuated pain responses in LPB neurons and elevated pain thresholds in pain states. 3) Mice with ablation of STN-LPB neurons developed pain-like behaviors after capsaicin injection or SNI surgery. These data suggest that STN neurons do not contribute to the development of pain states, but play important roles in modulating pain behaviors in pain states.

Our previous paper revealed that optogenetic excitation of STN terminals in different downstream brain areas such as the ventral pallidum, internal segment of the globus pallidus, and substantia nigra pars reticulata differentially reduces mechanical and thermal thresholds in physiological condition, while inhibition of individual terminals differently mitigates pain modalities in parkinsonian mouse models. It suggests that the pain modulatory effects of STN projection neurons may depend on their projections. In the present study, optogenetic modulation was restricted to LPB-projecting STN neurons or their projections to the LPB. Our data show that the pain modulatory effects of STN-LPB neurons and projections are different from those of

STN projections to other brain regions. These results extend our previous finding that STN neurons in different neural pathways may have different modulatory effects on pain modalities. However, further investigations are needed to clarify whether STN neurons projecting to different downstream regions are largely separated.

2. Related to these concerns, and in agreement with another reviewer, it might be advisable for the authors to parse the data into 2 or more manuscripts, with tighter rigor and rationale. In the process, moving much of the important data currently in the supplementary data section into the main manuscript might be beneficial.

Response: We agree with the reviewer that we include a large amount of data in this manuscript. However, it is a nice experience for us to work with leading experts in the field to complete a nice research. We confess that some of important data in the supplementary figures may be moved into the main figures. But considering the space limit and discernability of details in each panel, it is extremely difficult for us to incorporate more data in the main figures. Additionally, the journal discourages splitting up the manuscript. Therefore, we remain all data in this manuscript and show data more relevant to the main story in the main figures.

3. The "sticky tape" experiment does not address the concern that manipulating STN affects reflexive pain metrics.

Response: We conducted the adhesive tape removal test to address whether manipulation of neurons affects motor function being recruited in pain reflex because it is used in behavioral neuroscience to measure sensorimotor reflex and motor coordination (Liu et al., 2018; Schallert et al., 1982; Starkey et al., 2005). The adhesive removal and mechanical / thermal withdrawal may differ in appearance, but they may recruit similar physiological procedures. Removal of a sticky tape includes at least two steps: sensing the uncomfortable stimulation of a sticky tape on the hind limb (sensory procedure) and taking off the sticky tape from the hind limb (motor procedure). And

mechanical or thermal withdrawal includes two similar steps: sensing the unfavorable mechanical and thermal stimulation on the hind limb (sensory procedure) and withdrawing the hind limb to avoid of further stimulation (motor procedure). In sensory aspects, both the sticky tape and mechanical or thermal are uncomfortable stimulation and cause aversion. In motor aspects, both removal of sticky tape and withdrawal move the hind leg to terminate the uncomfortable feeling. Except for these similarities, removal of sticky tape involves the coordination of both hind legs. It is reasonable to postulate, to our understanding, that if STN manipulation does not affect the performance of mice in the adhesive tape removal test, it likely does not affect motor reflex in response to mechanical and thermal stimulation. There may be another test, beyond our knowledge, more accurately represents motor reflex involved in pain-like behaviors.

4. The absence of controls animals with CNO injections only, without DREADDs, continues to be a concern.

Response: We added a cohort of mice to address this issue. We injected AAV retro-hSyn-Cre in the STN (Day 0), AAV-EF1 α -DIO-mCherry in the SNr (Day 14), and AAV-CaMKII-GCaMP6f in the LPB (Day 14). Three to four weeks later, we performed fiber photometry to examine the responses of LPB neurons to pain stimulation after i.p. injection of saline or CNO. We found that CNO did not affect the responses of LPB neurons to pain stimulation. Because of the space limit, we show the data in Supplementary Figure 8.

5. "Page 6, line 140: Provide a more descriptive detail of the finding, beyond stating "affect the activity of LPB neurons". This prior concern is not addresses, as no quantification is provided.

Response: We quantified the strength of the STN-LPB projection in Fig 2K and Fig. 2M, and provide more detailed description in the Results.

6. *I do not understand the authors' response to reviewer 2 question regarding analysis of spontaneous rather than miniature synaptic events. "TTX does not affect both amplitude and frequency of sIPSCs and sEPSCs": What does this mean? "significant changes of amplitude and frequency of both sIPSCs and mIPSCs hint the involvement of postsynaptic and presynaptic mechanisms, respectively": This seems incorrect. "enhancement or inhibition of the activity of the presynaptic neurons leads to an increase or a reduction of sIPSC frequency"" this seems incorrect, too.*

Response: We appreciate the reviewer to point out these issues. We removed the data comparing sIPSCs in STN neurons between sham and SNI mice and replaced them with those comparing mIPSCs in STN neurons between sham and SNI mice.

References:

- Crossland, W. J., 1985. Anterograde and retrograde axonal transport of native and derivatized wheat germ agglutinin in the visual system of the chicken. *Brain Res* 347, 11-27.
- Goshgarian, H. G., Buttry, J. L., 2014. The pattern and extent of retrograde transsynaptic transport of WGA-Alexa 488 in the phrenic motor system is dependent upon the site of application. *J Neurosci Methods* 222, 156-164.
- Levy, S. L., White, J. J., Lackey, E. P., Schwartz, L., Sillitoe, R. V., 2017. WGA-Alexa Conjugates for Axonal Tracing. *Curr Protoc Neurosci* 79, 1 28 21-21 28 24.
- Liu, Y., Latremoliere, A., Li, X., Zhang, Z., Chen, M., Wang, X., Fang, C., Zhu, J., Alexandre, C., Gao, Z., Chen, B., Ding, X., Zhou, J. Y., Zhang, Y., Chen, C., Wang, K. H., Woolf, C. J., He, Z., 2018. Touch and tactile neuropathic pain sensitivity are set by corticospinal projections. *Nature* 561, 547-550.
- Mantyh, P. W., Peschanski, M., 1983. The use of wheat germ agglutinin--horseradish peroxidase conjugates for studies of anterograde axonal transport. *J Neurosci Methods* 7, 117-128.
- Menet, V., Prieto, M., Privat, A., Gimenez y Ribotta, M., 2003. Axonal plasticity and functional recovery after spinal cord injury in mice deficient in both glial fibrillary acidic protein and vimentin genes. *Proc Natl Acad Sci U S A* 100, 8999-9004.
- Schallert, T., Upchurch, M., Lobaugh, N., Farrar, S. B., Spirduso, W. W., Gilliam, P., Vaughn, D., Wilcox, R. E., 1982. Tactile extinction: distinguishing between sensorimotor and motor asymmetries in rats with unilateral nigrostriatal damage. *Pharmacol Biochem Behav* 16, 455-462.
- Starkey, M. L., Barritt, A. W., Yip, P. K., Davies, M., Hamers, F. P., McMahon, S. B., Bradbury, E. J., 2005. Assessing behavioural function following a pyramidotomy lesion of the corticospinal tract in adult mice. *Exp Neurol* 195, 524-539.
- Xu, X., Song, L., Kringel, R., Hanganu-Opatz, I. L., 2021. Developmental decrease of entorhinal-hippocampal communication in immune-challenged DISC1 knockdown mice. *Nat Commun* 12, 6810.

REVIEWERS' COMMENTS

Reviewer #1 (Remarks to the Author):

Thank you very much to the authors for the improvement of their manuscript and responses to my comments.

I have no more comments.

Reviewer #2 (Remarks to the Author):

The Authors have addressed all of my concerns.

Reviewer #3 (Remarks to the Author):

Unfortunately, the authors' responses do not allay concerns that I, and previous reviewers, raised previously.

- The authors' main conclusion regarding the role of the STN to PB pathway remains unsupported by the findings. They do confirm previous findings that PB is causally related to pain behaviors. They also show that manipulating STN inputs affects PB activity and pain behaviors. This is not surprising, as it is well known that manipulating *any* inputs to PB has similar effects. Their finding that STN to LPB inputs are altered in pain states is novel and interesting. However, the finding that ablating this pathway has no effect on pain behaviors argues against a causal role for this pathway in pain perception. The authors' argument that the results are consistent with a "modulatory" role for this pathway is not supported by the findings. I encourage the authors to present their interesting data, but to be less dogmatic in their interpretations.

- The relegation of key findings to Supplementary Data, to comply with space restrictions, remains problematic. Almost all of the "supplementary" data are essential for supporting the authors' conclusions, and should be presented within the body of the manuscript. Even if this requires separate manuscripts, or permission from the editors to exceed the page limits.

Reviewer #3 (Remarks to the Author):

Unfortunately, the authors' responses do not allay concerns that I, and previous reviewers, raised previously.

*1. The authors' main conclusion regarding the role of the STN to PB pathway remains unsupported by the findings. They do confirm previous findings that PB is causally related to pain behaviors. They also show that manipulating STN inputs affects PB activity and pain behaviors. This is not surprising, as it is well known that manipulating *any* inputs to PB has similar effects. Their finding that STN to LPB inputs are altered in pain states is novel and interesting. However, the finding that ablating this pathway has no effect on pain behaviors argues against a causal role for this pathway in pain perception. The authors' argument that the results are consistent with a "modulatory" role for this pathway is not supported by the findings. I encourage the authors to present their interesting data, but to be less dogmatic in their interpretations.*

Response: We agree with the reviewer that the STN-LPB pathway is not essential for the development of acute and persistent pain states because in mice subjected to ablation of STN-LPB neurons, capsaicin, CFA, and SNI still reduced pain thresholds.

As the reviewer mentions, we showed that the STN-LPB pathway is altered in acute and persistent pain states (Fig. 1, 8). Using optogenetic stimulation to mimic the hyperactivity of this pathway, we observed a significant reduction in mechanical and thermal thresholds in naïve mice (Fig. 4, 6, 7). These results hint that the potentiation of the STN-LPB pathway facilitates pain perception and may be involved in central sensitization in acute and persistent pain states.

In mice STN neurons were not ablated, we observed that pain threshold in acute and persistent pain states was increased by inhibition of STN-LPB neurons (Fig. 4) and the STN^{Glu}-LPB projection (Fig. 6, 7), and stimulation of the SNr^{GABA}-STN projection (Fig. 10, supplementary Fig. 10) as well.

Our above-mentioned bidirectional optogenetic manipulation data provide a set of evidence to support that the STN-LPB pathway plays a modulatory role in pain perception.

2. The relegation of key findings to Supplementary Data, to comply with space restrictions, remains problematic. Almost all of the "supplementary" data are essential for supporting the authors' conclusions, and should be presented within the body of the manuscript. Even if this requires separate manuscripts, or permission from the editors to exceed the page limits.

Response: We thank the reviewer to raise this problem. Because of this critique, the editor gives us permission to increase the number of main figures. In this revision, we reorganized the data and included as many data as possible into 10 main figures.